# A systematically-revised ribosome profiling method for bacteria reveals pauses at single-codon resolution

Fuad Mohammad[1], Rachel Green[1,2], Allen R Buskirk[1]*

[1]Department of Molecular Biology and Genetics, Johns Hopkins University School of Medicine, Baltimore, United States; [2]Howard Hughes Medical Institute, Johns Hopkins University School of Medicine, Baltimore, United States

**Abstract** In eukaryotes, ribosome profiling provides insight into the mechanism of protein synthesis at the codon level. In bacteria, however, the method has been more problematic and no consensus has emerged for how to best prepare profiling samples. Here, we identify the sources of these problems and describe new solutions for arresting translation and harvesting cells in order to overcome them. These improvements remove confounding artifacts and improve the resolution to allow analyses of ribosome behavior at the codon level. With a clearer view of the translational landscape in vivo, we observe that filtering cultures leads to translational pauses at serine and glycine codons through the reduction of tRNA aminoacylation levels. This observation illustrates how bacterial ribosome profiling studies can yield insight into the mechanism of protein synthesis at the codon level and how these mechanisms are regulated in response to changes in the physiology of the cell.
DOI: https://doi.org/10.7554/eLife.42591.001

*For correspondence:
buskirk@jhmi.edu

## Introduction

Local elongation rates vary considerably during protein synthesis depending on the codon, amino acid sequence, and mRNA structure. These variations can have dramatic effects on gene expression. Stretches of codons in leader peptides that are translated slowly under starvation conditions, for example, regulate the transcription of downstream biosynthesis genes (e.g. in the *E. coli trp* operon) (*Yanofsky, 1981*). In a similar manner, rare codons in structural genes have been implicated in fine-tuning translational rates in order to favor proper protein folding (*Kim et al., 2015*; *Kimchi-Sarfaty et al., 2007*; *Komar, 2009*; *Zhou et al., 2013*). The amino acid sequence of the nascent polypeptide can also alter elongation rates. Interactions between side chains and the exit tunnel can rearrange nucleotides within the peptidyl-transferase center, locking the ribosome in an inactive conformation (*Ito and Chiba, 2013*). For example, the *E. coli* SecM peptide uses stalling as a key feature in an elegant genetic switch in which stalling leads to changes in the local mRNA structure that promote translation of the downstream *secA* gene (*Nakatogawa and Ito, 2002*). Just as elongation can reshape mRNA structures, the converse is also true: strong mRNA structures can pause ribosomes and trigger rescue pathways associated with ribosome stalling and arrest (*Doma and Parker, 2006*). Although codon usage, amino acid sequence, and mRNA structure have each been shown to affect local elongation rates, our current understanding of these various features is insufficient to predict their effects on single genes, let alone across the genome.

In principle, ribosome profiling has the capacity to reveal pausing sites throughout the transcriptome with unprecedented clarity. In this approach, the positions of ribosomes on mRNAs are determined by sequencing ribosome-protected mRNA fragments (RPFs) (*Ingolia et al., 2009*); an increase in ribosome density at a site relative to its local context is evidence of a slower elongation rate

(*Ingolia et al., 2011*). In practice, however, reliably detecting pauses in ribosome profiling data from bacterial samples has been challenging because the methods used to arrest translation and harvest cells perturb the position of ribosomes, thus obscuring the in vivo translational landscape. Although these problems are not well characterized for the bacterial system, they have been carefully documented in several important studies in yeast. In the earliest yeast protocols, the eukaryotic elongation inhibitor cycloheximide (CHX) was added directly to cultures prior to harvesting the cells by centrifugation. It soon became clear that this method introduces several artifacts (*Gerashchenko and Gladyshev, 2014*). First, ribosome occupancy is enriched at the 5'-end of coding sequences and within upstream open reading frames because initiation continues during CHX treatment even though elongation is blocked. Second, because cycloheximide binding is reversible, when it falls off, ribosomes restart elongation only to be arrested again upon rebinding, blurring the position of the ribosome at the codon level (*Hussmann et al., 2015*). Finally, and perhaps most importantly, translational distress that is caused by cycloheximide may trigger a host of biological changes during the growth period, further obscuring the biology of interest in the samples. For these reasons, most researchers now prefer to harvest yeast by rapid filtration, adding cycloheximide to the lysis buffer only, not directly to growing cultures. Although it is difficult a priori to predict how ribosome density should look in vivo, the fact that yeast studies now report a negative correlation between codon usage and ribosome density (rare codons show higher levels of ribosome density consistent with the idea that they are decoded more slowly) argues that these improvements capture differences in elongation rates that were obscured by the earlier methods (*Hussmann et al., 2015*; *Weinberg et al., 2016*). With the additional refinements we recently reported for eukaryotic ribosome profiling, the negative correlation between codon usage and ribosome density is even more pronounced (*Wu et al., 2019*).

Although these improvements have made it possible to observe translational pauses with high resolution in yeast and other higher eukaryotes, these and other problems persist in bacterial ribosome profiling studies, limiting them to low resolution and masking the true in vivo translational landscape. To address these limitations, we have systematically optimized the ribosome profiling protocol to improve resolution in order to gain insights into the mechanism of protein synthesis in bacteria. Here, we report that the methods used to arrest translation and harvest cells are generally more problematic in bacteria than in yeast, blurring the signal and even inducing sequence-specific pauses. We developed new techniques to flash-freeze cultures and arrest translation robustly without the use of antibiotics. With these new methods, we obtain robust single-codon resolution in profiling samples from bacteria for the first time: for example, experimentally-induced pauses become crystal clear, appearing only in the ribosomal A site when decoding for a particular tRNA is rate-limiting, rather than blurring over several codons as before. We further found with this increased resolution that filtering *E. coli* cultures induces pauses at Ser and Gly codons as the corresponding tRNAs are no longer adequately aminoacylated. In describing our improvements to the profiling protocol, we survey the wide variety of methods used to generate profiling libraries from *E. coli*—unlike the case with yeast, no consensus has emerged to date—and make a case for what we believe to be 'best practices.' By bringing together these improvements in one place, we hope to help the bacterial research community to capitalize on the potential of ribosome profiling to yield insight into the molecular mechanisms of protein synthesis and the regulation of these mechanisms as a function of changes in the physiology of the cell.

## Results

### How to handle ribosomal footprints of various lengths

#### The 3'-end of footprints gives the best information about ribosome position

Ribosome profiling libraries in *S. cerevisiae* routinely show single-codon resolution and strong three nucleotide (nt) periodicity arising from the translocation of ribosomes along the mRNA one codon at a time (*Ingolia et al., 2009*). These features of the data are clearly evident in plots of average ribosome density across thousands of genes aligned at their start codons (*Figure 1A*, blue). In stark contrast, an equivalent plot with typical *E. coli* profiling data shows a diffuse peak at the start codon, roughly 20 nt wide, with no evidence of 3 nt periodicity arising from the reading frame of the ribosome (*Figure 1A*, *E. coli* (center), grey). The primary reason that the *E. coli* data are blurry is that

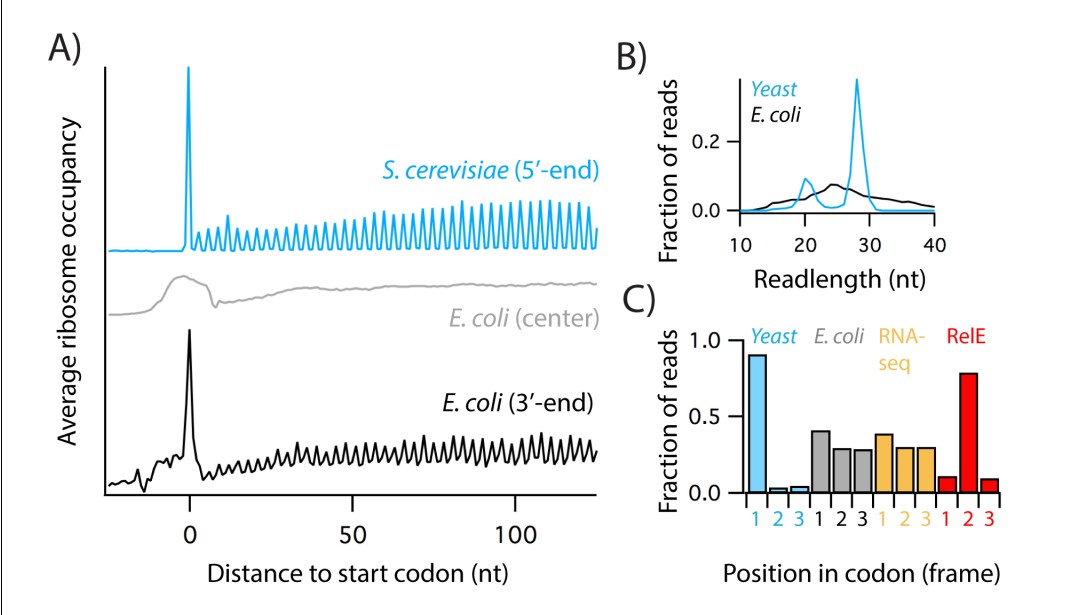

**Figure 1.** Comparison of ribosome profiling data from yeast and *E.coli*. (**A**) Average ribosome density on genes aligned at the start codon using the 5'-end of reads in yeast (library SRR1042864), or the center or 3'-end of reads from *E. coli* (library SRR1734438). (**B**) Length distribution of yeast and *E. coli* ribosome-protected fragments mapping uniquely to coding sequences. (**C**) The fraction of reads at the first, second, or third nt within codons in yeast profiling data (blue), *E. coli* profiling data (grey), RNA-seq from total RNA digested with MNase (yellow), and profiling data in which nucleases RelE and MNase were used to generate ribosome-protected footprints (red). See also *Figure 1—figure supplement 1*.

DOI: https://doi.org/10.7554/eLife.42591.002

The following figure supplement is available for figure 1:

**Figure supplement 1.** Preferential isolation of long RPFs increases ribosome density at SD-like motifs within open reading frames.

DOI: https://doi.org/10.7554/eLife.42591.003

bacterial ribosome protected mRNA fragments (RPFs) show a broad distribution in length, from 15 to 40 nt, whereas the majority of yeast RPFs are 28 nt long (*Figure 1B*). Because these 28 nt reads are fully trimmed by RNase I to the 5' and 3'-boundaries of the ribosome, the position of yeast ribosomes can be reliably determined from either end of the read. Faced with the difficulties posed by RPFs of variable lengths, the first bacterial profiling studies took an agnostic approach, distributing ribosome density broadly across the center of the reads (*Li et al., 2012*; *Oh et al., 2011*). Following this early protocol, most subsequent profiling studies in *E. coli* have used this center-assignment strategy, immediately limiting the resolution of the data. There is a better way: we and others found that the position of the ribosome can be more accurately inferred from the 3'-end of bacterial RPFs (*Balakrishnan et al., 2014*; *Nakahigashi et al., 2014*; *Woolstenhulme et al., 2015*). In data analyzed this way, the start codon peak is only 1–2 nt wide (*Figure 1A*, black). This higher resolution allows us to identify the codon positioned in the ribosomal A site so that analyses of pausing during starvation or upon depletion of translation factors can be performed at the codon level with greater precision (*Woolstenhulme et al., 2015*). Although it is tempting to interpret the three nt periodicity in these data as evidence of reading frame, as observed in yeast (*Figure 1A*, blue), this periodicity arises from the specificity of the nuclease used to generate RPFs, and not from the ribosome, as discussed below.

## On the sequence specificity of nucleases

Bacterial studies (*Li et al., 2012*; *Oh et al., 2011*) use MNase to generate RPFs because RNase I, the nuclease used in yeast and many higher organisms, is an *E. coli* enzyme and is inhibited by *E. coli* ribosomes (*Kitahara and Miyazaki, 2011*). (In our hands, even 10,000 units of RNase I, an entire tube, did not collapse *E. coli* polysomes to monosomes on a sucrose gradient). Unfortunately, unlike RNase I, MNase shows significant sequence specificity (*Dingwall et al., 1981*). Due to this specificity,

and to a lesser extent the ligases used in generating cDNA libraries, ribosome profiling data exhibit a high level of noise. The peaks in ribosome density across a single gene may vary in height by 1000-fold or more. For some datasets, nucleotide bias at the 5'- and 3'-ends of RPFs accounts for more of this variation than the identity of the codon in the A site of the ribosome (*O'Connor et al., 2016*). To minimize the effects of these artifacts, we average ribosome density over thousands of instances of a site of interest (such as Pro codons) (*Woolstenhulme et al., 2015*). We also routinely compare ribosome profiling data to total RNA-seq samples prepared using the same protocol (*Hwang and Buskirk, 2017*; *Mohammad et al., 2016*). In addition, several sophisticated computational methods have been developed to minimize the effects of cloning bias (*O'Connor et al., 2016*) and adjust the ribosome density by taking into account the specificity of MNase (*Zhao et al., 2018*). In analyzing ribosome profiling data, care must be taken to avoid mistaking technical artifacts for real biology.

As an example of the problems that sequence bias can cause, consider the following: *E. coli* data show a modest three nt periodicity (*Figure 1A*, black) that could easily be misinterpreted as the movement of ribosomes along transcripts one codon at a time. Unlike the yeast data, where > 90% of reads align to the first nt of codons (*Figure 1C*, blue), however, the periodicity in E. coli is quite weak: 40% of reads align to the first nt, 30% to the second, and 30% to the third (*Figure 1C*, grey). In previous work, we showed that the periodicity in the *E. coli* data primarily arises from artifacts of the nuclease digestion and not from the reading frame of the ribosome (*Hwang and Buskirk, 2017*). MNase cleaves more efficiently before A and T. Because codons are used at different frequencies, A and T occur more often than expected at the second nt of codons in the *E. coli* genome. Together, this bias in the genome and the sequence specificity of MNase yield the weak periodicity seen in ORFs in *Figure 1A* (black) and 1C (grey). We verified this hypothesis in an earlier study (*Hwang and Buskirk, 2017*), finding that total RNA-seq samples prepared by MNase digestion show the same weak periodicity exclusively in ORFs even in the absence of intact ribosomes (*Figure 1C*, yellow).

For studies where the reading frame of the ribosome is essential, such as studies of programmed frameshifting, we reported that generating RPFs with the endonuclease RelE can reveal three nt periodicity in *E. coli* profiling samples (*Hwang and Buskirk, 2017*) (*Figure 1C*, red). This is because RelE only cleaves mRNA inside the ribosome, precisely after the second nt of the A-site codon (*Pedersen et al., 2003*). On the other hand, RelE also shows strong sequence specificity (*Hwang and Buskirk, 2017*), and because we assign ribosome positions from the 3'-end of ribosome footprints, this specificity introduces bias at the A-site codon that interferes with analyses of pausing. In contrast, because MNase cleaves at the 3'-boundary of the ribosome, roughly 12 nt away from the A site, its sequence selectivity creates little or no bias at the A site when averaged over many instances of a codon of interest. In short, RelE yields excellent reading frame, and so is useful for analyses of frameshifting and stop-codon readthrough, but because it interferes with pausing analyses, we continue to use MNase in the studies described below.

## On the importance of isolating all ribosomal footprint lengths

Why do bacterial RPFs vary so much in length? Some have argued that the sequence specificity of MNase prevents it from fully digesting back to the ribosome boundaries. While MNase is partially responsible for the poor three nt periodicity of bacterial ribosome profiling data, it cannot be blamed for the large differences in read lengths. MNase does reliably degrade mRNA to within 1–2 nt of the 3'-boundary of the ribosome; this is why 3' alignment of RPFs is so good at revealing codon resolution. While this small amount of variability does interfere with analyses of reading frame, it does not provide an explanation for the broad distribution in RPF length observed in *E. coli* samples (*Figure 1B*). Another finding exculpating MNase is that yeast profiling libraries generated with MNase show a tight distribution of RPFs centered at 28 nt very much like those generated with RNase I (*Gerashchenko and Gladyshev, 2017*). An important clue to the heterogeneity of bacterial RPFs comes from our unpublished studies in *B. subtilis*. There, we generated ribosome profiling libraries using RNase I, which is not inhibited by *B. subtilis* ribosomes, and observed very broad read length distributions similar to those obtained from *E. coli* (data not shown). These observations argue that the factor responsible for the heterogeneity of RPF lengths in bacteria is not the nuclease, but instead something inherent to bacterial ribosomes.

Eukaryotic ribosomes protect different lengths of mRNA from nuclease digestion at different states in the translation cycle. For example, elongating ribosomes primarily yield 28 nt RPFs when

CHX is used to arrest translation whereas terminating ribosomes (with stop codons in their A sites) yield RPFs one nt longer due to changes in the mRNA conformation induced by release factors (*Brown et al., 2015*; *Ingolia et al., 2011*). Ribosomes trapped at the end of truncated mRNA species (generated by nuclease activity in the cell) yield 16 nt RPFs that have yielded much information about various mRNA decay processes (*Guydosh and Green, 2014*; *Guydosh et al., 2017*; *Guydosh and Green, 2017*). In addition to the well-characterized 28 nt RPFs, elongating ribosomes also generate 21 nt RPFs in eukaryotes (*Figure 1B*) (*Lareau et al., 2014*). Recent studies indicate that 21 nt RPFs arise from mRNA cleavage by RNase I within the ribosome just downstream of the A-site codon when the A site is devoid of tRNA (*Wu et al., 2019*). In contrast, ribosomes in the hybrid or pre-translocation state still carry tRNA in the 40S A site that blocks RNase I activity within the ribosome, yielding 28 nt RPFs trimmed to the ribosome boundaries. These differences in footprint size in yeast will be powerful clues that allow researchers to determine the specific functional state of the ribosome at a given site.

Much less is known about the sources of variability in footprint length in bacterial studies. Given that the distribution of RPF lengths is very wide and that we do not fully understand why, it is troubling that there is no consensus in bacterial studies about the size of RPFs that should be isolated and sequenced, as can be seen in 25 representative libraries from different labs shown in *Figure 2* (*Baggett et al., 2017*; *Burkhardt et al., 2017*; *Haft et al., 2014*; *Latif et al., 2015*; *Li et al., 2014*; *Li et al., 2012*; *Liu et al., 2013*; *Marks et al., 2016*; *Oh et al., 2011*; *Subramaniam et al., 2014*). The earliest bacterial studies (*Li et al., 2012*; *Oh et al., 2011*) isolated RPFs 28–42 nt in length (e.g. libraries L9-10, L18-19) and many later studies followed this early protocol, preferentially isolating long RPFs (L21-L24), whereas others have preferentially isolated shorter RPFs (L15-17). We argue that the best approach is to cast a wide net, isolating all potentially relevant RPFs, 15–40 nt in length. In our hands and in others (*Li et al., 2014*), such a preparation yields a broad distribution of RPFs with a peak at 24 nt (e.g. libraries L1 and L20).

Casting a wide net to isolate the entire footprint distribution is essential to ensure an accurate representation of ribosomes in various stages of the translational cycle. In an earlier study, we showed, for example, that 70S ribosomes on start codons yield RPFs that are significantly longer (30–35 nt) than RPFs from elongating ribosomes in open reading frames (*Mohammad et al., 2016*). It is possible that the presence of initiation factors in newly assembled 70S complexes results in longer RPFs, but we favor a model in which direct mRNA-rRNA interactions protect additional mRNA from digestion by MNase. As expected from such an interaction between the Shine-Dalgarno motif and the anti-SD sequence in 16S rRNA, the extra length in RPFs at start codons is found at the 5'-end of the read. This explains why the 5'-ends of RPFs are more variable and why assigning the ribosome position by the 3'-end of the read is more precise. Consistent with this hypothesis, RPFs observed on SD-like motifs within open reading frames are also longer on average (*O'Connor et al., 2013*). At these internal sites, far from start codons, ribosomes should no longer be bound to initiation factors, thus arguing that the mRNA-rRNA base-pairing is primarily responsible for protecting mRNA and yielding longer RPFs, as discussed previously (*Mohammad et al., 2016*).

This effect of mRNA-rRNA base pairing on footprint length explains the early observation that SD-like motifs in open reading frames induce ribosome pausing. In one of the first bacterial ribosome profiling papers (*Li et al., 2012*), strong enrichment of ribosome density led to the conclusion that SD-like motifs are the main source of translational pauses in bacteria. We argued (*Mohammad et al., 2016*) that this observation arose from a biased sampling of the relevant RPFs: long footprints were selectively isolated in the original study (L18), yielding a positive correlation between ribosome density and the strength of internal SD-like motifs (*Li et al., 2012*), but in other libraries where short footprints were selectively isolated (e.g. L17), there is actually a negative correlation (*Figure 1—figure supplement 1*). Given that RPFs that base pair with rRNA tend to be longer (30–35 nt) (*O'Connor et al., 2013*), isolating only longer RPFs leads to artificial enrichment of ribosome density at SD-like motifs (*Mohammad et al., 2016*). Pauses at SD-like motifs are not observed in our libraries, including new ones prepared with the improvements described below (*Figure 1—figure supplement 1*), nor were they detectable in various biochemical assays (*Borg and Ehrenberg, 2015*; *Chadani et al., 2016*; *Mohammad et al., 2016*). These data suggest that SD pauses are an artifact of the method, highlighting the importance of isolating an unbiased population of RPFs.

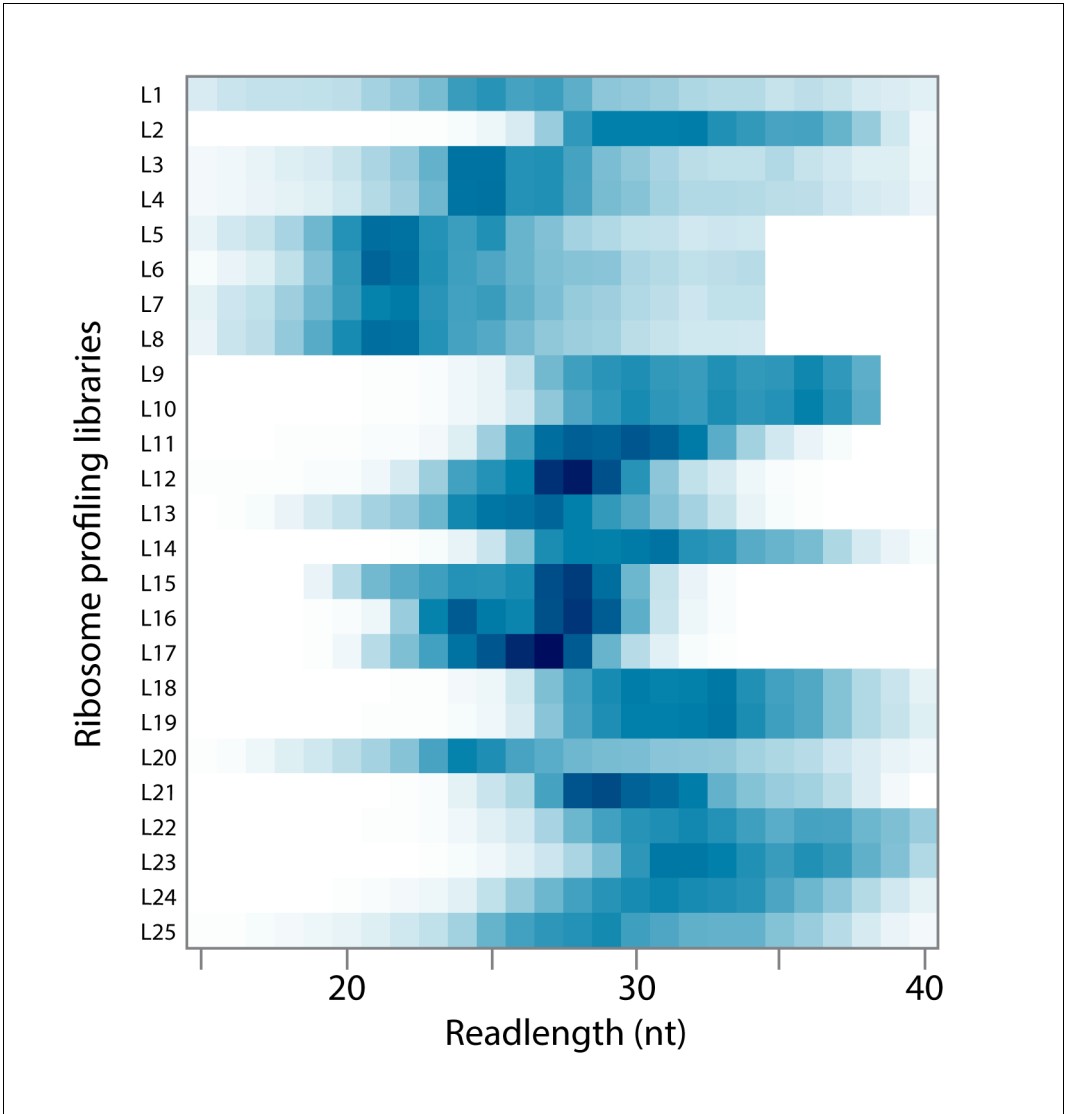

**Figure 2.** Heat map of the distribution of read lengths in published *E.coli* ribosome profiling libraries from several labs. See *Figure 2*, *Figure 2—source data 1* for details.

DOI: https://doi.org/10.7554/eLife.42591.004

The following source data is available for figure 2:

**Source data 1.** Table of ribosome profiling libraries with references and accession numbers.

DOI: https://doi.org/10.7554/eLife.42591.005

## How to stop translation

### Chloramphenicol in the media induces artifacts at the gene level

A critical consideration in the preparation of ribosome profiling libraries is how ribosomes are trapped to most accurately preserve their position during harvesting, cell lysis, and mRNA digestion. In the same way that CHX was initially added to yeast cultures to arrest translation prior to harvesting cells by centrifugation (*Ingolia et al., 2009*), some bacterial studies have followed a similar strategy, adding the elongation inhibitor chloramphenicol (Cm) to the growing culture (L1-L10, *Figure 3A*). As was observed (*Gerashchenko and Gladyshev, 2014*) in yeast grown in CHX, the addition of Cm to bacterial cultures skews the translational profiles because initiation continues even as the antibiotic arrests elongation, leading to an accumulation of ribosome density at the 5'-end of ORFs. This can be seen when we compute asymmetry scores for each gene by taking the log2 of the

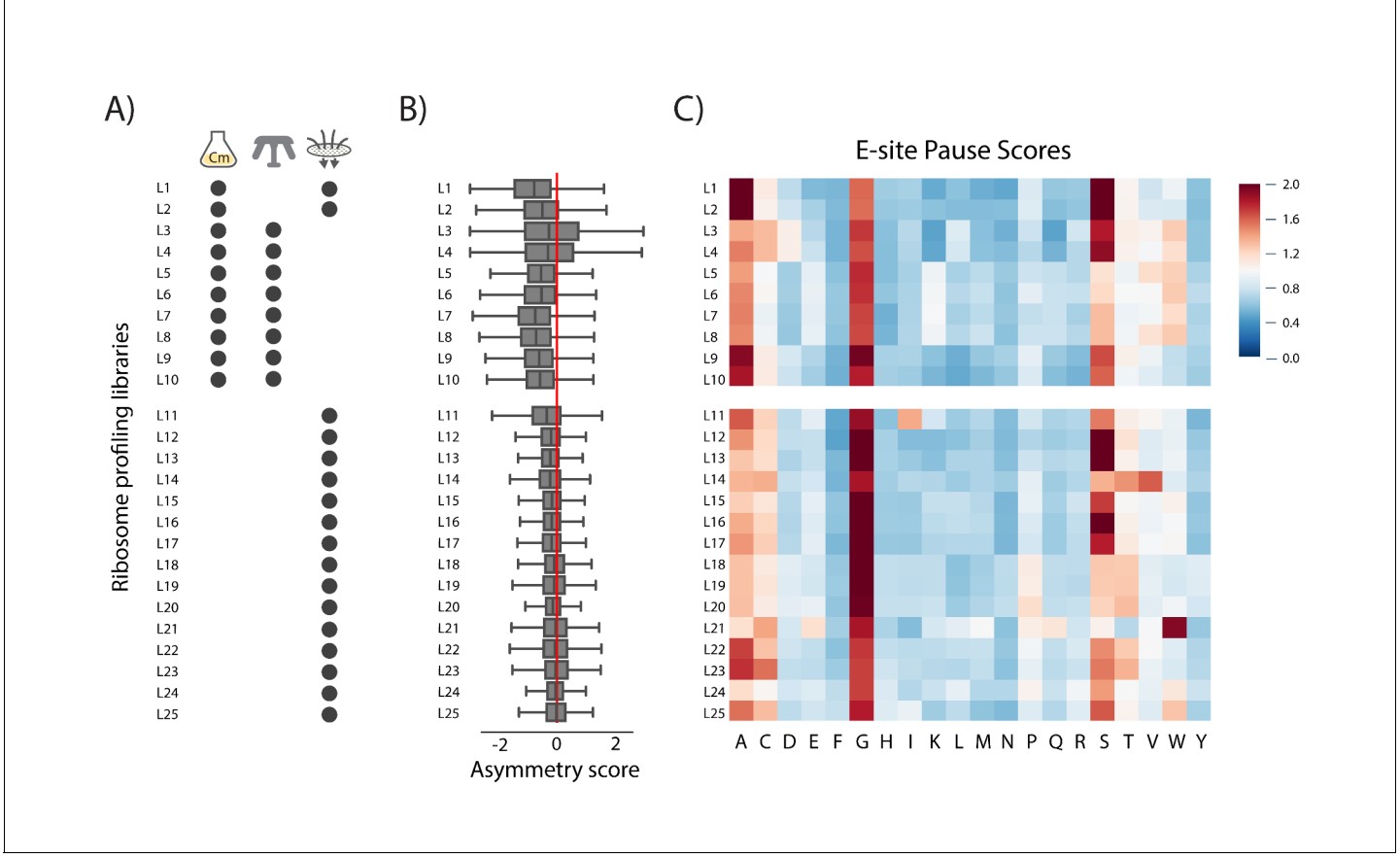

**Figure 3.** Chloramphenicol (Cm) alters ribosome density at the gene and codon level in published *E.coli* ribosome profiling libraries. (**A**) Cultures are harvested by centrifugation or filtration. L1-L10 were treated with Cm in the media prior to harvesting; all samples were prepared with Cm in the lysis buffer. (**B**) Distribution of asymmetry scores, the log2 value of the number of reads in the second half of a gene divided by the number of reads in the first half. Genes with more ribosomes at the 5'-end than the 3'-end have negative values. (**C**) Heat map of pause scores for the codon in the ribosomal E site (corresponding to the penultimate amino acid in the nascent peptide). See also *Figure 3—figure supplement 1*.

DOI: https://doi.org/10.7554/eLife.42591.006

The following figure supplement is available for figure 3:

**Figure supplement 1.** Treating cultures with Cm prior to harvesting skews estimates of protein synthesis levels in different ways depending on the gene length.

DOI: https://doi.org/10.7554/eLife.42591.007

ratio of ribosome density in the second half of the gene over the density in the first half; using this metric, genes with more ribosomes in the first half yield negative values. The distribution of asymmetry scores for thousands of genes shown in *Figure 3B* reveals that ribosome density is strongly enriched in the first half of the majority of genes in samples where Cm is added to the media (L1-L10). This artifact affects genes differently depending on their length, artificially inflating the number of reads per kilobase per million mapped reads (RPKM) for short genes and reducing RPKM values for long genes (*Figure 3—figure supplement 1*). It particularly complicates estimates of the amount of translation of leader peptides and other non-canonical sites (*Gerashchenko and Gladyshev, 2014*). Simply comparing experimental and control cultures that are both treated with antibiotics does not resolve these issues. Control samples may be affected by antibiotics differently from experimental samples harboring a mutation or growing under different conditions, even if the antibiotic treatment and downstream sample handling steps are identical.

To avoid these artifacts, protocols now recommend collecting cells by rapid filtration and freezing in liquid nitrogen, thus arresting translation by freezing cells rather than by adding antibiotics prior to centrifugation (L11-L25, *Figure 3A*) (*Becker et al., 2013*). Even in the first bacterial ribosome

profiling study (*Oh et al., 2011*), the authors observed that filtration removed Cm-induced distortions in the data. Our standard protocol has been to pour 200 mL of a culture that has reached $OD_{600}$ = 0.3 into a vacuum filtration apparatus; as the cells accumulate on the membrane, and before the media has completely passed through, we scrape the cells off and plunge them into liquid nitrogen. Because this procedure is quick (about 30 s of filtration), we have hoped that translation is not disrupted prior to freezing. The rapidly frozen samples are then cryogenically lysed in a cryo-mill together with frozen lysis buffer; the buffer contains Cm so that translation will not resume in the lysate as the samples are thawed and processed. In libraries from many labs prepared in this fashion (L11-L25), the asymmetry scores are close to zero (*Figure 3B*), meaning that ribosome density is evenly distributed across each gene rather than being enriched near the 5'-end. The mean asymmetry scores from these libraries were significantly less skewed than those from the Cm pre-treated libraries L1-L10 (two-tailed, independent t-test, p=$1.7 \times 10^{-7}$). As observed in yeast, rapid filtering and freezing provides the means to harvest cultures without pre-treatment with antibiotics, eliminating a host of associated problems.

## Chloramphenicol also induces artifacts at the codon level

While rapid filtration and freezing allow Cm to be eliminated from growing cultures, protocols still include Cm in the lysis buffer to prevent ongoing translation. Although Cm has been widely assumed to be a general translation inhibitor, recent ribosome profiling and toeprinting studies have revealed that Cm inhibits translation in a sequence-specific manner that alters the pausing landscape at the codon level (*Marks et al., 2016*; *Mohammad et al., 2016*; *Nakahigashi et al., 2014*; *Orelle et al., 2013*). These studies have shown that the ability of Cm to inhibit the peptidyl-transfer reaction depends on which amino acids are in the nascent chain, especially the identity of the penultimate residue. In profiling data, Cm-dependent pauses are observed at the codon in the ribosomal E site (which encodes the penultimate residue). These effects are quantitated as pause scores for each amino acid in the E site, calculated by taking the ribosome density at the appropriate codons divided by the average density on a given gene; as such, these values reflect an average of individual pause scores for thousands of sites (an average or 'meta' analysis). As expected based on the earlier studies, in samples where Cm is added to the media to arrest translation (L1-10), we observe strong pauses when Ala, Gly, and Ser codons are positioned in the E site (*Figure 3C*). These pauses make sense structurally because Cm binds within the active site of the ribosome, blocking peptidyl transfer (*Dunkle et al., 2010*). Having small side chains in the residues of the nascent chain near the active site likely facilitates Cm binding and activity.

We also calculated pause scores for libraries from several different labs to see what effect Cm has when present only in the lysate to prevent ongoing translation (L11-25). Although the intensity varies, these libraries all show E site pausing at the same codons (Ala, Gly, and Ser) as observed in samples where Cm was added to the media (L1-10). Despite the clear improvements that came with rapid filtration and freezing, we were disappointed to find that even when Cm is only present in the lysis buffer, there remained Cm signatures that seemed likely to obscure the visualization of naturally-occurring ribosome pauses. These data led us to hypothesize that translation in the lysate leads to these Cm-specific translational pauses.

## Measuring and arresting translation in lysates

To investigate the extent to which ribosomes continue to translate in the lysate during the preparation of ribosome profiling libraries, we developed a biochemical assay to measure the amount of newly-synthesized protein (*Figure 4A*). In this assay, we add [14C]Lys-tRNA to frozen lysates directly after cells are cryogenically pulverized, allowing them to react in translational elongation as the lysates thaw for 15 min according to our usual protocol (in the presence or absence of any inhibitors). Translational activity is then revealed by the incorporation of [14C]-Lys into TCA-precipitable nascent peptides. As a control, we add the same charged [14C]Lys-tRNA to lysate that has been heat-killed at 90°C for 10 min such that the translational machinery is fully denatured. Indeed, we observed very little [14C]-Lys incorporation in our boiled control when compared to active lysate with no antibiotics, which yielded robust incorporation into TCA-precipitable protein (No drug, *Figure 4B*). These data indicate that ribosome profiling lysates synthesize proteins robustly in the absence of any added translational inhibitors. Importantly, when the standard 1 mM Cm was added

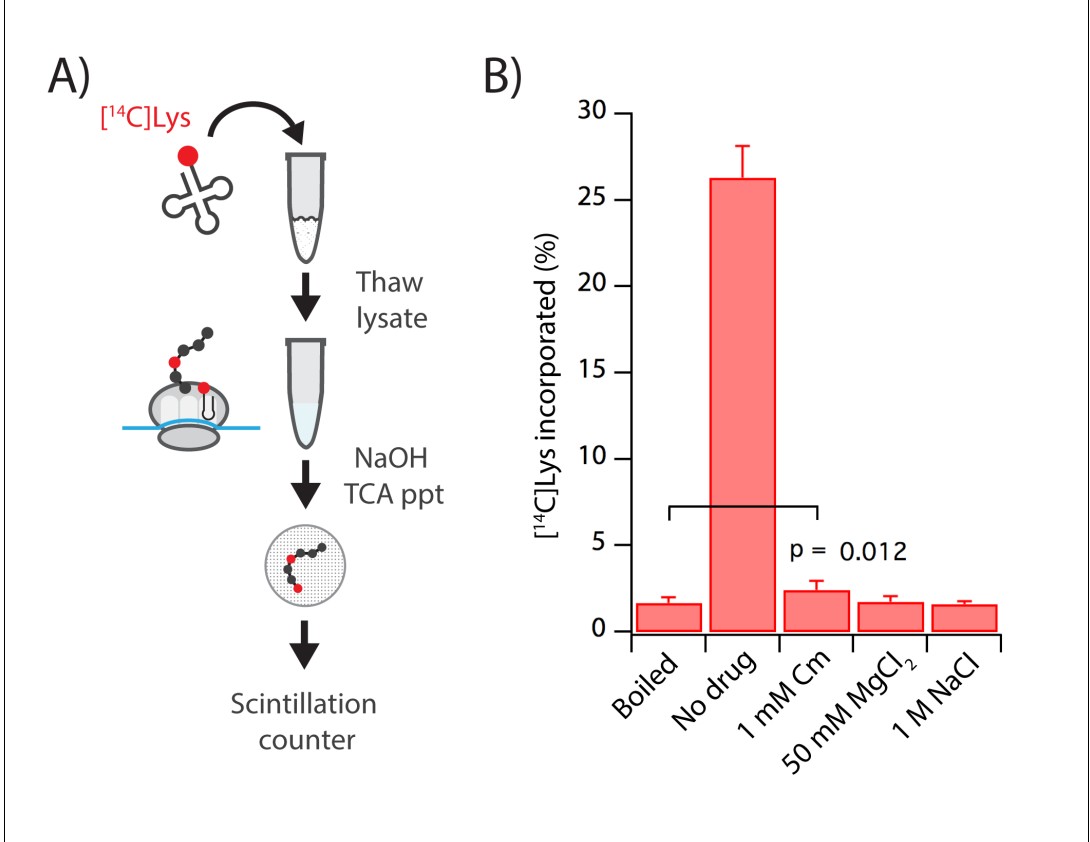

**Figure 4.** High salt buffers arrest translation after cell lysis better than Cm. (**A**) We added [14C]Lys-tRNA$^{Lys}$ to frozen lysate that was then thawed for 15 min. [14C]Lys that was incorporated into nascent peptides can be selectively precipitated with TCA after tRNA hydrolysis under alkaline conditions. (**B**) Lysates were made with buffers containing 1 mM Cm, 50 mM $MgCl_2$, or 1M NaCl. The boiled sample was denatured prior to addition of Lys-tRNA. Error bars reflect the standard deviation of four technical replicates. The boiled and Cm samples were compared using a one-tailed, paired t-test.
DOI: https://doi.org/10.7554/eLife.42591.008

to the lysate, we observed a small but statistically significant increase in TCA-precipitable signal compared to the denatured control (*Figure 4B*). While this amount of translation activity is modest, this result, taken together with the sequence-specific pauses observed in ribosome profiling data, suggests that chloramphenicol imperfectly blocks translation when added to the lysis buffer, leading to the sequence-specific pausing that we observe in ribosome profiling libraries.

We next used the [$^{14}$C]Lys-incorporation assay to identify alternatives to Cm that might more effectively arrest translation in lysates. To avoid issues with antibiotic specificity, we turned to observations made in the early days of in vitro ribosome biochemistry characterizing the sensitivity of translation extracts to mono- and divalent salts. We found that concentrations of $MgCl_2$ higher than 50 mM inhibit the incorporation of [$^{14}$C]Lys, yielding a signal that is statistically indistinguishable from background levels (*Figure 4B*). Similarly, we found that 1 M NaCl robustly inhibited translation in the lysate. We suspect that these conditions hinder conformational changes essential for ribosomes to undergo the various steps of elongation.

## Ribosome profiling with high salt buffers improves the resolution of pauses

We next sought to incorporate these new lysis buffers into our ribosome profiling protocol. We cryogenically lysed cells in buffers containing high $MgCl_2$ or high NaCl and then, because MNase is incompatible with these buffers, we pelleted ribosome complexes over a sucrose cushion and resuspended them in the standard lysis buffer prior to digestion (*Figure 5—figure supplement 1A*). By pelleting ribosome complexes, we effectively deplete nucleotides and translation factors so that antibiotics or high salt concentrations are no longer necessary to arrest translation. Before

proceeding to the digestion step, we ran the samples over a sucrose gradient to confirm that the combined steps of high salt lysis and ribosome pelleting did not reduce the integrity of polysomes. For this analysis, we quantitated the ratio of polysomes to monosomes and ribosome subunits for each sample; a reduction in this ratio may be due to cleavage of the mRNA or subunit splitting. We found that after pelleting, a sample with the standard Cm-containing buffer showed a reduced ratio (2.2) compared to a non-pelleted sample (2.7, *Figure 5—figure supplement 1B*). After pelleting, the 50S peak increased, suggesting that some ribosomes split into subunits, while the 30S peak disappeared, as 30S subunits do not pellet through the cushion. The 1 M NaCl buffer appears to slightly worsen this effect; in pelleted samples, the polysome ratio with the 1M NaCl buffer is 2.0 compared to 2.2 for the standard Cm-containing buffer. In contrast, the high $MgCl_2$ buffer (we now use 150 mM) promotes polysome stability: it has the highest polysome ratio (2.9), showing the least amount of subunit splitting after pelleting (*Figure 5—figure supplement 1B*). These results indicate that high $MgCl_2$ buffer may be optimal for ribosome profiling because of its ability to prevent translation in the lysate while still maintaining polysome integrity.

We next prepared ribosome profiling libraries using both conditions (either 150 mM $MgCl_2$ or 1 M NaCl in the lysis buffer) to arrest translation. As expected based on the in vitro translation assays (*Figure 4*), in libraries prepared with these buffers, the codon-specific pauses induced by Cm in the E site are no longer present. When Cm is added only in the lysis buffer in library L26, E-site Gly and Ser pauses are observed (*Figure 5A*, left, first column). Importantly, in libraries L27 and L28 prepared with the high $MgCl_2$ and NaCl buffers respectively, the Gly pauses in the E site are greatly reduced and the Ser pauses completely disappear (*Figure 5A*, left, second and third columns).

With this methodology that effectively stops translation in lysates, we next deliberately induced pauses at specific sites to see whether the resolution improves at these known, biologically relevant pauses. First, we treated cells with mupirocin, an inhibitor of isoleucyl-tRNA synthetase (*Hughes and Mellows, 1978*), anticipating that as Ile-tRNA levels drop, ribosome density would increase specifically at Ile codons. In a library prepared in the traditional manner using Cm in the lysate to arrest translation (L32), in addition to the strong pause at Ile codons in the A site, we see that ribosome density is enriched as far as three codons downstream (*Figure 5B*, black). This observation is readily explained by ongoing translation in the lysate in the presence of Cm that blurs the pausing signal at Ile codons. In contrast, in library L30 prepared using the high $MgCl_2$ buffer, the downstream pauses disappear and the pause at the A site is higher because translation is truly arrested (*Figure 5B*, blue). In both libraries, density is enriched about 25 nt upstream of the Ile codons as the next ribosome on the mRNA stacks behind the ribosome paused at the Ile codon. Taken together, these data show that the high $MgCl_2$ buffer not only removes the pauses induced by the E-site specificity of Cm in the lysate, it also sharpens resolution at genuine pauses by fully arresting translation in the lysate.

Returning to libraries L27 and L28 prepared with the high $MgCl_2$ and high NaCl buffers from untreated cells, the clarity these buffers bring to our data also reveals pauses that were missed in earlier studies. We observe strong pauses at Ser codons now in the ribosomal A site and, to a lesser extent, pauses at Gly codons there as well (*Figure 5A*, right). In plots of average ribosome occupancy at Ser codons, density is only enriched in the A site (*Figure 5C*, red and blue). In libraries prepared traditionally using Cm in the lysate (e.g. L26), the same pause is spread over the A, P, and E sites due to ongoing translation in the lysate (*Figure 5C*, black). Note that the Ser and Gly pauses in the A site shown here are distinct from the Ala, Ser, and Gly pauses observed in the E site in samples pre-treated with Cm in the media (L1-L10, *Figure 3C*). A-site pauses are usually the result of defects in decoding whereas E-site pauses (discussed above) arise from the effects of the nascent chain on Cm's ability to inhibit peptidyl transfer. In earlier studies, both of these effects are in play: some combination of ongoing translation and the sequence specificity of Cm generates the complex pausing landscapes seen in L11-L25 (*Figure 3C*).

## Filtering cultures induces A-site Gly and Ser pauses

Initially we were puzzled by the observation of strong pauses at Gly and Ser codons in the A site. These pauses suggest that Gly and Ser codons are decoded slowly as the ribosome waits for the appropriate aminoacyl-tRNA to bind and react with the nascent peptide chain. A-site pauses have been observed when cells are starved for specific amino acids in yeast (*Guydosh and Green, 2014*; *Lareau et al., 2014*) and in *E. coli* as shown with mupirocin above. However, it seemed unlikely that

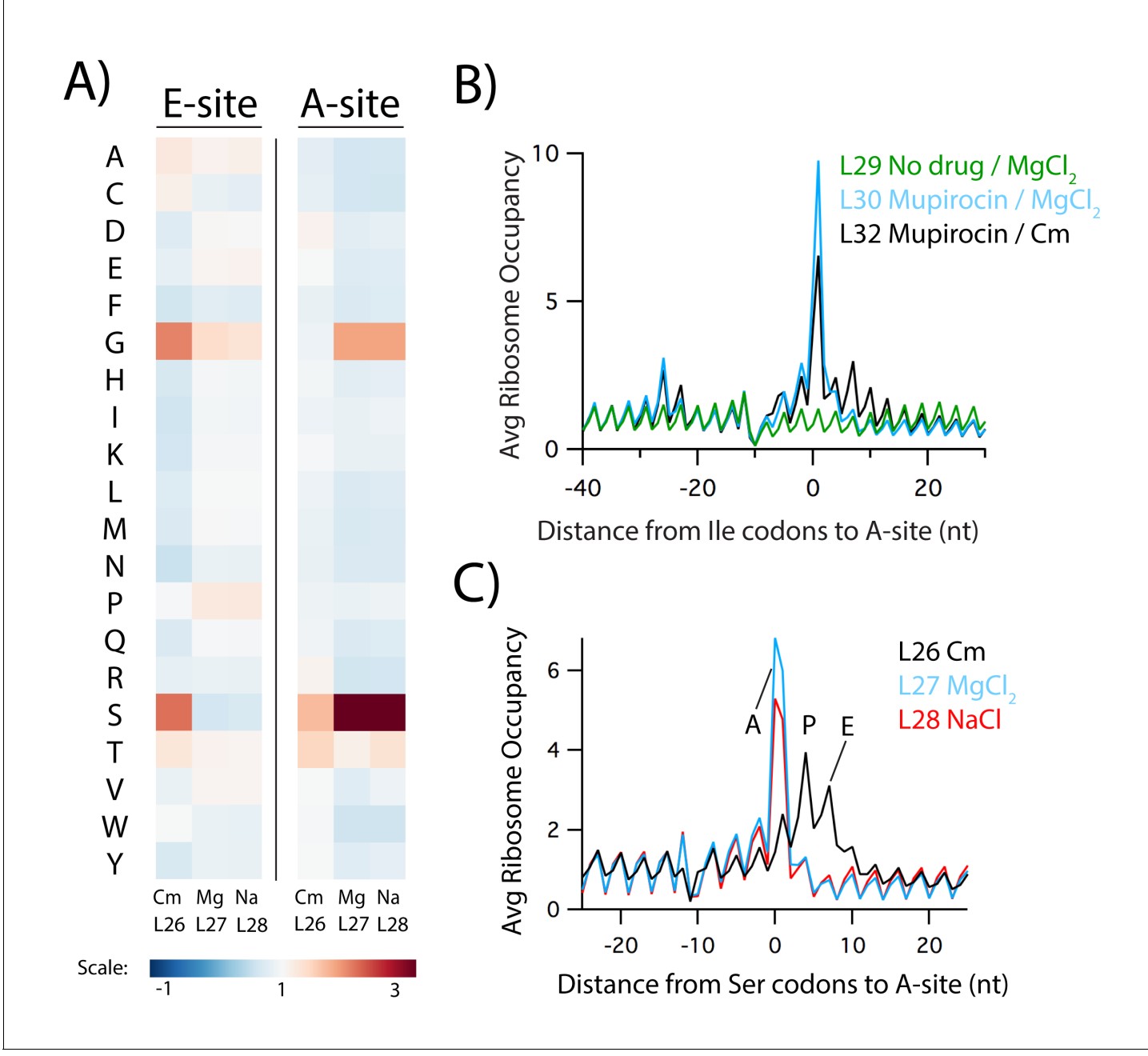

**Figure 5.** Pausing is crystal clear in samples prepared with high salt buffers instead of Cm. (**A**) Heatmap of pause scores for codons for all 20 amino acids in either the E or A site of the ribosome from samples prepared with lysis buffers containing Cm, 150 mM $MgCl_2$, or 1 M NaCl (libraries L26, L27, and L28 respectively). (**B**) Average ribosome occupancy aligned at Ile codons for samples treated with mupirocin, an inhibitor of Ile-tRNA synthetase, and an untreated control (**L29**), using lysis buffers with either high $MgCl_2$ or Cm (L30 and L32 respectively). (**C**) Average ribosome occupancy aligned at Ser codons in untreated cells using lysis buffers containing Cm, high $MgCl_2$, or high NaCl (libraries L26, L27, and L28 respectively).

DOI: https://doi.org/10.7554/eLife.42591.009

The following figure supplement is available for figure 5:

**Figure supplement 1.** Incorporating high salt lysis buffers into ribosome profiling.

DOI: https://doi.org/10.7554/eLife.42591.010

cells in our samples were starving for Ser and Gly since the cultures were grown in a complete MOPS media supplemented with all 20 amino acids, including 10 mM Ser, and were harvested in early log phase ($OD_{600}$ = 0.3) before nutrients are significantly depleted. Moreover, Ser and Gly pauses in the A site were not reduced in libraries prepared using other amino-acid-rich media formulations (data not shown). We wondered if the Ser and Gly pauses we observe might be caused not by low levels of available nutrients, but from the way that we harvest cells to prepare ribosome profiling libraries.

An important clue to the source of these pauses comes from the pattern of ribosome occupancy downstream of Ser codons in these datasets. In heat maps of ribosome density aligned at Ser codons, we observed strong pauses in the A site followed by a region of reduced ribosome density, regardless of the lysis buffer used (*Figure 6A*). What is striking is that the level dips for 10–15 codons after the pause but then returns to a higher level further downstream. What explains this unusual pattern? Under optimal conditions, there is a steady state level of ribosome density across messages as most of the ribosomes elongate with roughly similar rates (*Figure 6B*). If a pause is induced that is strong enough to become rate-limiting, it impedes the progress of upstream ribosomes while downstream ribosomes continue elongating and are released at stop codons. The quality control machinery likely removes paused ribosomes from the mRNA, further lowering the density downstream of the pause site (*Subramaniam et al., 2014*). Eventually, the system will come to a new steady state in which the ribosome density between the pause site and stop codon will be lower than it was in the initial steady state, but relatively constant across the downstream ORF. When cells are treated for 10 min with mupirocin (L30), for example, ribosome density is depleted downstream of the strong pauses at Ile codons compared to a control (L29, *Figure 6C*). Importantly, the level of ribosome density downstream of Ile codons is uniformly lower (extending to the stop codon) indicating that a new steady state has been reached.

In contrast, the dip in ribosome density following Ser pauses extends for only 10–15 codons (*Figure 6A*), suggesting that the system has not yet reached steady state, as though a pause has been induced just prior to translational arrest and library preparation. We reasoned that this might mean that Ser pauses in this case reflect an artifact of the method and not a true depiction of the translational landscape in the exponentially-growing culture. We wondered if acute Ser pauses arise as cells are harvested by filtration just before translation is fully arrested when the cells are frozen in liquid nitrogen.

While it is challenging to come up with a method of harvesting cells without either filtration or centrifugation, we decided that rather than concentrating cells prior to lysis, we would concentrate ribosomes after lysis by pelleting them over a sucrose cushion. We developed a new approach in which we spray 50 mL of culture directly into liquid nitrogen to form small, frozen drops that are then ground in a cryo-mill together with 10X lysis buffer. To test this method, we harvested 50 mL of culture directly in liquid nitrogen and 200 mL from the same culture with the standard filtering protocol. Both samples were prepared using the high $MgCl_2$ lysis buffer and pelleting over a sucrose cushion to remove the high salt concentrations that preclude efficient MNase digestion (*Figure 6D*). In plots of average ribosome density aligned at Ser codons, we see a strong A-site pause in the standard filtered sample (L29), as described above, but a complete loss of the Ser pause in the direct-freeze sample (L33, *Figure 6E*). The more modest A-site Gly pauses also disappear (see *Figure 7A* below). We conclude that filtration induces pausing at Ser and Gly codons.

## Filtration lowers the levels of aminoacylation of tRNA$^{Ser}$ and tRNA$^{Gly}$

Why does rapid filtering lead to ribosome pausing? Given that Gly and Ser pauses occur in the ribosomal A site and reflect reduced decoding rates, we asked if filtration lowers the aminoacylation levels of tRNAs encoding Ser and Gly. We extracted total tRNA from cells collected by either filtration or direct freezing. As a control, a fraction of each tRNA sample was pre-treated with mild base to deacylate all the tRNAs. We used periodate oxidation and β-elimination to distinguish between charged and uncharged tRNAs in these four samples. As uncharged tRNAs are selectively oxidized by periodate, after β-elimination they end up one nt shorter then charged tRNAs, allowing us to resolve the two species by PAGE and northern blotting using tRNA-specific probes. For the filtered sample, most of the tRNA$^{Ser}$ is uncharged, similar to the pre-treated, deacylated control, whereas in the direct-freeze sample, most of the tRNA$^{Ser}$ is aminoacylated and therefore one nt longer. These

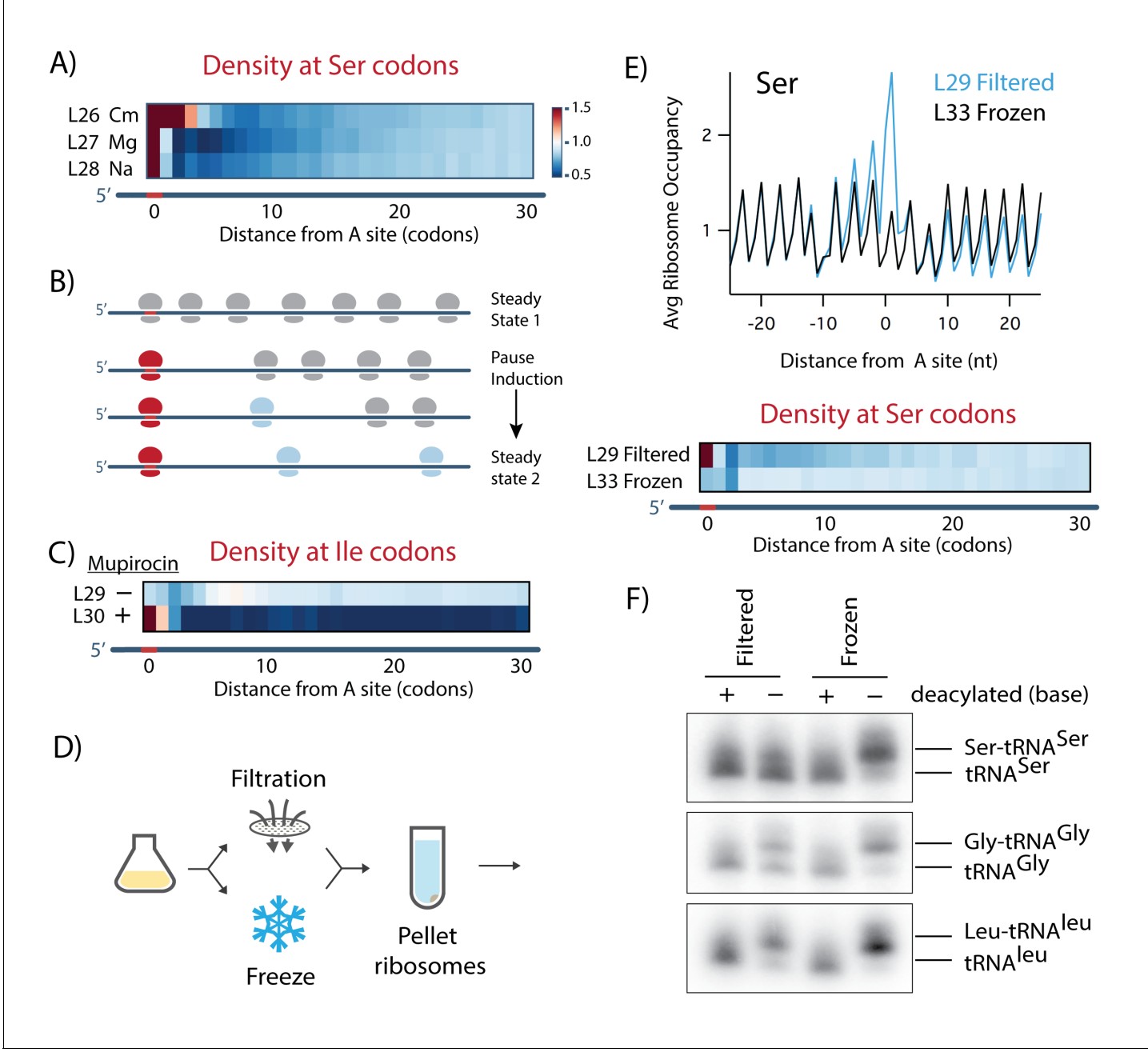

**Figure 6.** Filtering cells leads to ribosome pausing at Ser codons due to reduced levels of aminoacylated tRNA$^{Ser}$. (**A**) Heatmap of ribosome density downstream of Ser codons in samples harvested by filtration using lysis buffers containing Cm, 150 mM MgCl$_2$, or 1 M NaCl (L26, L27, and L28, respectively). (**B**) Model of how pausing affects ribosome density. Downstream of a pause site (shown in red), ribosomes continue elongation and are released at stop codons, such that downstream density drops until a new steady state is reached. (**C**) Heatmap of ribosome density downstream of Ile codons in untreated cells (L29) and after 10 min of mupirocin treatment (L30). (**D**) Schematic of the method used for panels 6E and 6F: from a single culture, samples were harvested by rapid filtration or by directly freezing the culture. Ribosomes were then pelleted through a sucrose cushion. (**E**) Plots and heatmaps of average ribosome density aligned at Ser codons in untreated cells that were filtered (L29) or frozen (L33). (**F**) Northern blot of Ser, Gly, and Ile tRNAs after periodate oxidation and β-elimination, a treatment that removes the final nt of tRNAs that are not charged. As a negative control, an aliquot of tRNA from filtered or frozen samples were pretreated in alkaline conditions to deacylate tRNA.

DOI: https://doi.org/10.7554/eLife.42591.011

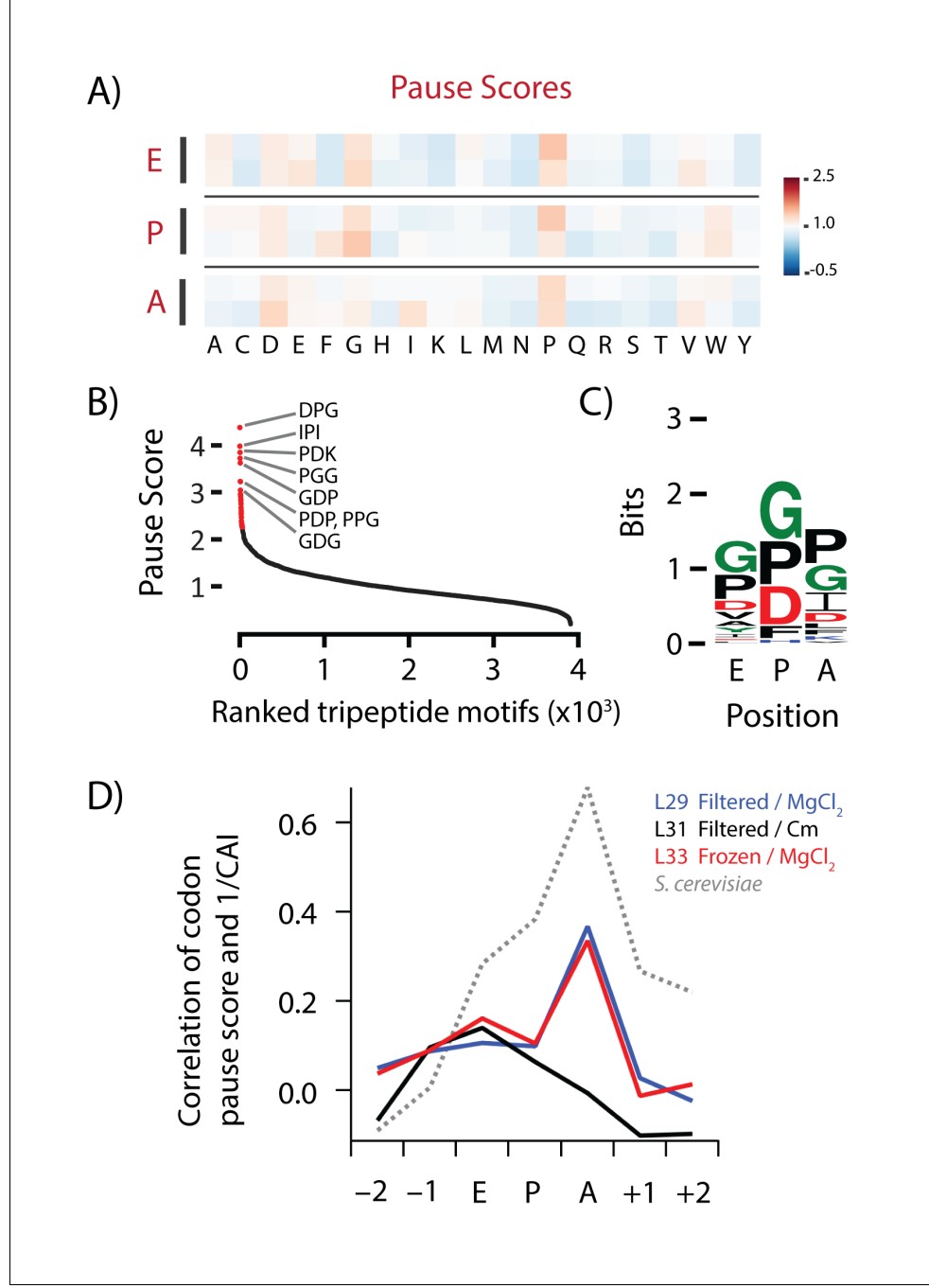

**Figure 7.** Samples harvested by direct freezing and lysed in high MgCl$_2$ buffer reveal subtle ribosome pauses that reflect known biology, pauses at polyproline motifs and at rare codons. (**A**) Heatmap of pause scores in two biological replicates harvested by direct freezing (L33 and L35). (**B**) Ranking of all tripeptide motifs by their pause scores with the motif occupying the A, P and E sites of the ribosome in library L35. (**C**) Sequence logo of the top 50 tripeptide motifs from 7B. (**D**) Spearman correlation between ribosome density at each codon and the inverse value of its codon-adaptation index (CAI), a measure of codon usage and optimality. The correlation was calculated for codons within the ribosome (E, P, and A-site codons) and two codons on either side. The *E. coli* data are from libraries L29 (filtered, MgCl$_2$), L31 (filtered, Cm), and L33 (frozen, MgCl$_2$) and the *S. cerevisiae* data are from SRR1049521 (***Subtelny et al., 2014***).
DOI: https://doi.org/10.7554/eLife.42591.012

data show that filtering reduces the level of aminoacylation of tRNA$^{Ser}$. A similar effect, albeit more modest, is observed for tRNA$^{Gly}$; perhaps half of the tRNA$^{Gly}$ is uncharged in the filtered sample while it is fully charged in the direct-freeze sample. The higher fraction of charged tRNA$^{Gly}$ (compared to tRNA$^{Ser}$) is consistent with the observation that Gly pauses are generally weaker than Ser pauses in the ribosome profiling data. Importantly, filtration did not have a discernible impact on the charging levels of a control tRNA, tRNA$^{Leu}$, as expected, given that pauses were not observed at Leu codons. Together, these northern blots provide a clear explanation for the origins of pausing at Ser and Gly revealed in the ribosome profiling data; in cells harvested by filtration, there is a reduction in the aminoacylation levels of tRNA$^{Ser}$ and tRNA$^{Gly}$.

## The pausing landscape in samples prepared by direct freezing reflects real biology

Compared to centrifugation and filtration, the direct freezing method that we have developed yields libraries with a dramatically different translational pausing landscape. Without the strong pauses at Gly and Ser codons that are induced by Cm in the E site or by filtering in the A site, we now are able to see generally weak pauses at Pro, Asp, and Gly codons (*Figure 7A*, L33 and L35). Pro exhibits the most significant pausing in all three of the ribosomal tRNA-binding sites (A, P, and E). Pro is well known to inhibit peptidyl transfer when found at the last two residues in nascent polypeptides (corresponding to the pauses at the E and P site codons) (*Doerfel et al., 2013*; *Tanner et al., 2009*; *Wohlgemuth et al., 2008*; *Woolstenhulme et al., 2013*). Pro also is a poor peptidyl acceptor, likely explaining the pause in the A site as well (*Pavlov et al., 2009*). Looking at all three codons in the E, P, and A sites together, we found that the tripeptides with the highest pause scores contained permutations of Asp, Gly, and Pro (e.g. DPG, PGG, PPG, *Figure 7B*); a sequence logo in *Figure 7C* showing the information content in the top 50 motifs reveals enrichment of these amino acids in all three sites. The fact that these pauses reflect known limitations of the translational machinery at Pro codons suggests that we are at last looking at an in vivo pausing landscape that is no longer masked by artifacts of the profiling method.

Another expectation is that rare codons will be decoded more slowly than abundant codons and therefore have higher levels of ribosome density. Indeed, it has been known for decades that highly expressed genes in *E. coli* tend to avoid rare codons (*Plotkin and Kudla, 2011*; *Sharp and Li, 1987*), arguably because these codons are read slowly by tRNAs that are present at low concentrations (*Bulmer, 1991*; *Ikemura, 1981*). A variety of metrics have been developed to calculate the codon usage of genes (CAI) (*Sharp and Li, 1987*) or its adaptation to the levels of tRNA (tAI) (*dos Reis et al., 2004*). In early ribosome profiling papers in yeast (*Charneski and Hurst, 2013*; *Qian et al., 2012*) and *E. coli* (*Li et al., 2012*), little correlation was observed between ribosome density and these metrics. Recently, it was found that the addition of cycloheximide to the media created artifacts that masked what was in fact a substantial correlation (*Hussmann et al., 2015*). For example, in *S. cerevisiae* data prepared in the absence of cycloheximide (*Subtelny et al., 2014*), there is a reasonable correlation between ribosome density and 1/CAI in the A site (*Figure 7D*) (*Weinberg et al., 2016*). Reassuringly, the correlation in the E and P sites (and nearby codons) is lower. In *E. coli* data obtained with the standard Cm buffer, the expected correlation is not observed in the A site (L31, black), presumably because translation continues in the lysate. Importantly, however, in our new samples prepared with the high MgCl$_2$ buffer (L29 and L33), a significant correlation with 1/CAI is observed (Spearman $\rho$ = 0.36) and the correlation is highest in the A site, where decoding takes place. Again, it is essential to fully arrest translation in order to obtain the most physiologically relevant biological results.

## Discussion

Unlike ribosome profiling in yeast, there has been little consensus regarding the best practices for generating ribosome profiling libraries in bacteria. One of the challenges in working with bacteria is the broad distribution in the length of ribosome-protected footprints (RPFs). Although the majority of footprints are ~24 nt in length, we observe RPFs ranging from 15 to 40 nt. We argue that the lengths of ribosome footprints in bacteria are inherently more variable. In particular, unlike eukaryotic ribosomes, bacterial ribosomes can base-pair directly with mRNA in an interaction that

resembles Shine-Dalgarno/rRNA pairing during initiation, effectively yielding longer footprints wherever G-rich sequences are encountered. Although not demonstrated in bacteria, it may also be true that classical/hybrid conformations of the ribosome will yield different RPF lengths, as they do in yeast (*Lareau et al., 2014*; *Wu et al., 2019*). We recommend isolating a broad range of footprints (15–40 nt) in order to capture the entire distribution and to avoid introducing biases that may confound downstream analyses. And with this broad distribution of read lengths, we find that assigning the position of the ribosome using the 3'-end of footprints yields higher resolution than the center-assignment strategy, because most of the length heterogeneity is found at the 5'-end of reads (*Woolstenhulme et al., 2015*). For experiments where the reading frame of the ribosome is critical, addition of the nuclease RelE to the digestion reaction generates precise 3'-ends that yield strong three nt periodicity (*Hwang and Buskirk, 2017*), although this comes at the cost of sequence bias that interferes with analyses of pausing. Taken together, these insights and improvements make the broad distribution in footprint sizes manageable and dramatically improve the resolution of profiling data.

A second challenge for ribosome profiling studies is to develop methods to harvest cells and arrest translation without perturbing the in vivo translational landscape. As reported in yeast, treatment of cultures with antibiotics prior to cell lysis can distort calculations of the number of ribosomes per message and the signal from non-canonical initiation sites (*Gerashchenko and Gladyshev, 2014*). Whenever possible, treatment of cultures with antibiotics should be avoided. This limitation effectively rules out centrifugation, which relies on antibiotics in the media to arrest translation during this lengthy procedure. The more commonly used method of rapid filtration (without antibiotics in the culture) can be a more effective strategy, especially when the primary goal is to count the number of ribosomes per message to determine differences in protein synthesis levels between two biological samples (*Figure 5—figure supplement 1C and D*).

For studies of mechanisms of protein synthesis and their regulation, however, the rapid filtering method and the standard Cm-containing lysis buffer are inappropriate because they impact ribosome density at the codon level. First, we found that because of Cm specificities imposed on translation through the penultimate amino acid, there is ongoing translation in the lysate during sample preparation leading to pauses at Ala, Gly, and Ser codons in the E site (*Marks et al., 2016*); these pauses are observed in essentially all published ribosomal profiling datasets (*Figure 3C*). We further discovered that filtration is problematic because it introduces pauses at Ser and Gly codons in the A site. To circumvent these issues, we made two substantive changes to the standard protocol which effectively allow translation to be arrested without introducing sequence-specific pauses: first, we directly freeze cultures in liquid nitrogen (avoiding the filtration step) and we use lysis buffers with >50 mM $Mg^{2+}$.

With these pausing artifacts removed, we capture a cleaner representation of ribosome density at the codon level and begin to glimpse pausing that authentically characterizes the in vivo translational landscape. First, we see subtle pauses with Pro codons in the three active sites of the ribosome, similar to those seen in cells lacking EFP, a elongation factor that promotes peptide-bond formation on these challenging substrates (*Doerfel et al., 2013*; *Peil et al., 2013*; *Ude et al., 2013*; *Woolstenhulme et al., 2015*). This observation suggests that even in wild-type *E. coli*, EFP may not be able to fully alleviate pausing at Pro codons sequences. Asp and Gly residues also appear to be translated slowly, particularly in combination with Pro residues, as previously seen in ribosome profiling data from yeast cells in which eIF5A is depleted (*Schuller et al., 2017*). Like EFP in bacteria, eIF5A alleviates pausing at polyproline stretches in eukaryotes (*Gutierrez et al., 2013*). Pro and Asp codons are enriched at sites of ribosome pausing in wild-type mammalian cells (*Ingolia et al., 2011*). These similarities suggest that all ribosomes struggle with pauses at these codons, probably due to slow rates of peptidyl transfer that result from the unusual limitations of the prolyl amino acid side chain.

Our new data also reveal a negative correlation between ribosome density and codon-adaptation index (CAI), consistent with the expectation that rare codons will be decoded by lower-abundance tRNAs more slowly than more abundant codons. Given the strong evidence of natural selection acting on codon usage in *E. coli* and *S. cerevisiae*, this result has long been expected, but this relationship was not revealed in early profiling studies. We now know that using antibiotics to arrest translation skews the position of ribosomes on messages to obscure the enrichment of ribosome density at non-optimal codons (*Hussmann et al., 2015*). Interestingly, the correlation that we

observe for *E. coli* is not as strong as for *S. cerevisiae* (*Weinberg et al., 2016*). One reason may be that the ribosomes are not all trapped at the same step in the translational cycle. Pauses on Pro codons suggest that some ribosomes are trapped during peptidyl transfer, whereas pauses on rare codons suggest that others are trapped with empty A sites during decoding. It is even possible that certain amino acid combinations are problematic for translocation. The signals from these different subsets of ribosomes interfere with each other. In yeast, it is now possible to use multiple antibiotics targeting different steps to tease apart these different states of elongating ribosomes. For example, analysis of populations of ribosomes arrested in the decoding step only reveals very high correlations between ribosome density and codon optimality metrics like CAI and tAI (*Wu et al., 2019*). A similar strategy may also improve these correlations in bacteria in the future. The methods described here are an important first step towards this goal, enabling for the first time studies of local elongation rates and their effects on protein folding or gene expression.

Already the clarity brought by our new methods has revealed a possible link between physiological stress and local translational rates: filtering cultures for as little as 30 s leads to ribosome pausing during the decoding of Ser and Gly codons. We confirmed that these pauses are caused by a sharp drop in the level of aminoacylation of these tRNAs triggered by the filtration. Consistent with these findings, Ignatova and co-workers reported A-site Ser pauses in *E. coli* profiling data and used tRNA microarrays to show that tRNA$^{Ser}$ has very low charging levels even in cultures grown in rich media (*Avcilar-Kucukgoze et al., 2016*). They found that these pauses do not resolve upon addition of more glucose or serine to the media. Likewise, our data show that A-site Ser and Gly pauses are induced by the methods used to harvest the cells, not by depletion of nutrients from the media. The fact that some published profiling datasets show A-site Ser pauses while others do not can probably be explained by subtle differences in harvesting procedures.

The nature of the stress induced by filtering remains unclear. It is not the change in temperature: strong Ser and Gly pauses are observed whether filtration is performed at room temperature, as usual, or in a warm room at 37°C (data not shown). The trigger could be the contact between a cell and the membrane, the contact between cells as they accumulate over time, or the mechanical stress as they are scraped from the membrane. We cannot rule out the possibility that as cells accumulate on the membrane, they are no longer able to take up Ser and Gly that are otherwise available in the media. Alternatively, Ser may be channeled away from protein synthesis for other purposes. Ser is used in many biochemical reactions, primarily as a donor of one-carbon units (through the folate cycle) for the biosynthesis of nucleotides and other amino acids (Gly, Thr, Met) (*Stauffer, 2004*). Bacteria deplete extracellular Ser faster than any other amino acid (*Selvarasu et al., 2009*), perhaps because they express Ser deaminases that convert Ser to pyruvate and ammonia. It has been proposed that the reason for this apparently wasteful reaction is that high levels of Ser are toxic to *E. coli* cells (*Zhang and Newman, 2008*). Regulation of intracellular Ser concentrations is therefore essential to balance its many roles in metabolism. Working out the mechanism for the reduction of charged tRNA$^{Ser}$ and tRNA$^{Gly}$ may yield insight into how cells regulate the flux of these important molecules at the center of so many metabolic pathways.

We are intrigued by the possibility that other physiological stresses may impact protein synthesis and vice versa. For example, when *B. subtilis* cells are cultured in media that induces biofilm formation, ribosomes pause at Ser codons, leading to the reduction of translation of an important transcription factor, SinR (*Subramaniam et al., 2013*). As the level of SinR drops, biofilm-related genes are no longer repressed, and cells switch to a program of matrix gene expression and biofilm formation. With the methods that we report here, we will be able to observe the effects of stress on the local translation rates across the genome, perhaps discovering similar phenomena relevant to other important physiological stresses in bacteria.

# Materials and methods

**Key resources table**

| Reagent type (species) or resource | Designation | Source or reference | Identifiers | Additional information |
|---|---|---|---|---|

*Continued on next page*

*Continued*

| Reagent type (species) or resource | Designation | Source or reference | Identifiers | Additional information |
|---|---|---|---|---|
| Strain, strain background (*Escherichia coli*) | E. coli MG1655 | E. coli genetic stock center | CGSC #6300 | |
| Sequence-based reagent | tRNASer northern probe | IDT | | GATTCGAACTCTGGAACCCTTT CGGGTCGCCGGTTTTC |
| Sequence-based reagent | tRNAGly northern probe | IDT | | GAATCGAACCCGCATCATCAGCTTGG |
| Sequence-based reagent | tRNALeu northern probe | IDT | | GACTTGAACCCCCACGTCCGTAA GGACACTAACACCTG |
| Sequence-based reagent | NEB Universal miRNA Cloning Linker | New England BioLabs | Cat# S1315S | 5' rAppCTGTAGGCACCATCAAT–NH2 3' |
| Sequence-based reagent | RT Primer | IDT | | /5Phos/AGATCGGAAGAGCGTCGTGTA GGGAAAGAGTGTAGATCTCGGTGGTCGC /iSP18/CACTCA/iSp18/TTCAGACGTGTG CTCTTCCGATCTATTGATGGTGCCTACAG |
| Sequence-based reagent | PCR Primer Forward | IDT | | AATGATACGGCGACCACCGAGATCTACAC |
| Sequence-based reagent | PCR Primer Reverse | IDT | | CAAGCAGAAGACGGCATACGAGATNNN NNNGTGACTGGAGTTCAGACGTGTGC TCTTCCG |
| Peptide, recombinant protein | Nuclease S7 (MNase) | Millipore Sigma | Cat# 10107921001 | |
| Peptide, recombinant protein | T4 polynucleotide kinase | New England BioLabs | Cat# M0201S | |
| Peptide, recombinant protein | T4 RNA Ligase 2, truncated | New England BioLabs | Cat# M0242S | |
| Peptide, recombinant protein | SuperScript III | ThermoFisher Scientific | Cat# 18080085 | |
| Peptide, recombinant protein | CircLigase ssDNA Ligase | Lucigen | Cat# CL4115K | |
| Commercial assay or kit | Ribo-Zero rRNA Removal Kit (Bacteria) | Illumina | Cat# MRZB12424 | |
| Chemical compound, drug | Mupirocin | Millipore Sigma | Cat# M7694 | |
| Chemical compound, drug | Chloramphenicol | Millipore Sigma | Cat# C0378 | |
| Chemical compound, drug | Transfer Ribonucleic Acid: Lysine specific | Chemical Block Ltd | NA | |
| Other | MOPS EZ Rich Defined Medium Kit | Teknova | Cat# M2105 | (Growth Media) |

## Bacterial culture conditions and lysis

*E. coli* MG1655 cells were grown overnight at 37°C in MOPS EZ Rich Defined media (Teknova) supplemented with 0.2% glucose and diluted 1:100 into 300 mL fresh media and grown at 37°C to an optical density of 0.3. Cultures were treated with 200 µM mupirocin (MPC, Sigma) or 1 mM chloramphenicol (Cm, Sigma) when indicated in the text. Cells were harvested either by filtration or by direct freezing of the culture in liquid nitrogen. Biological replicates consist of cultures from individual colonies grown on separate days.

Filtration was performed using a Kontes 90 mm filtration apparatus with 0.45 µm nitrocellulose filters (Whatman); cells were scraped from the filter before the media runs dry and were then frozen in liquid nitrogen. 0.65 mL of frozen lysis buffer was added to the pellets as indicated in the text. The standard lysis buffer is 20 mM Tris pH 8.0, 10 mM MgCl$_2$, 100 mM NH$_4$Cl, 5 mM CaCl$_2$, 0.1% NP-40, 0.4% TritonX-100, and 100 U/mL DNase I (Roche). To this buffer, 1 mM chloramphenicol, 1 M NaCl or 150 mM MgCl$_2$ was added as indicated in the text. The cells were cryogenically pulverized using

a Spex 6870 freezer mill with 5 cycles of 1 min grinding at 5 Hz and 1 min cooling. Lysates were thawed at room temperature and gently homogenized by passing through a 20 gauge syringe five times. Lysates were clarified by centrifugation at 20,000 g for 10 min at 4°C. For buffer exchange, 25 AU of RNA in the lysates was layered on top of a 1 mL sucrose cushion (20 mM Tris pH 7.5, 500 mM NH$_4$Cl, 0.5 mM EDTA, 1.1 M sucrose) and ribosomes were pelleted by centrifugation using a TLA 100.3 rotor at 65,000 rpm for 2 hr. Pellets containing ribosomes were re-suspended using resuspension buffer (0.2 mL of 20 mM Tris pH 8.0, 10 mM MgCl$_2$, 100 mM NH$_4$Cl, 5 mM CaCl$_2$, 0.1% NP-40, 0.4% TritonX-100) and used for subsequent experiments.

For samples harvested by direct freezing, 50 mL of culture at OD$_{600}$ of 0.3 was directly sprayed from a pipette into liquid nitrogen. The frozen culture was cryogenically pulverized together with 5.6 mL 10x lysis buffer (1x concentrations listed above) with 10 cycles of 1 min grinding at 8 Hz and 1 min cooling. Lysates were thawed at room temperature and pelleted over a 3 mL sucrose cushion (1.1 M sucrose, 20 mM Tris pH 8, 500 mM NH$_4$Cl, 10 mM MgCl$_2$, 0.5 mM EDTA) using a Ti-70 rotor at 70,000 rpm for 2 hr. Ribosome pellets were re-suspended in 200 µL resuspension buffer.

## Ribosome profiling library preparation

Lysates were processed for Illumina high throughput sequencing as follows: 18 AU of RNA was digested with 1,500 U of MNase (Nuclease S7, Roche) for 1 hr at 25°C and then quenched with EGTA at a final concentration of 6 mM. Samples were layered on a 10–50% sucrose gradient buffered with 20 mM Tris pH 8.0, 10 mM MgCl$_2$, 100 mM NH$_4$Cl and 2 mM DTT. Monosomes were isolated from the gradient after centrifugation in a SW41 rotor at 35,000 rpm for 2.5 hr at 4°C. 0.75 mL acid phenol pH 4.5 was added to 1 mL of the monosome fraction and incubated at 65°C for 5 min, followed by a second extraction with 0.75 mL acid phenol and finally with 0.6 mL chloroform before precipitating with isopropanol. 10 µg of RNA fragments were resolved by running samples on a 15% TBE Urea gel alongside size markers and 15–45 nt fragments were gel purified. Eluted RNA was then isopropanol precipitated and subsequently treated with T4 polynucleotide kinase (NEB) to dephosphorylate 3′ ends. After another round of isopropanol precipitation, the RNA fragments were ligated to the linker 5′ rAppCTGTAGGCACCATCAAT–NH2 3′ (NEB Universal miRNA Cloning Linker) using T4 RNA ligase (NEB) for 3 hr at 37°C. Ligated RNA fragments were resolved on 10% TBE Urea gels and gel extracted. Following another precipitation, rRNA fragments were subtracted using the Ribo-Zero rRNA removal kit for bacteria (Illumina). Ligated fragments were then reverse transcribed using SuperScript III (Invitrogen) at 48°C for 30 min, using the primer/5Phos/AGATCGGAAGAGCG TCGTGTAGGGAAAGAGTGTAGATCTCGGTGGTCGC/iSP18/CACTCA/iSp18/TTCAGACGTGTGC TCTTCCGATCTATTGATGGTGCCTACAG. Template RNA was degraded by adding 180 mM NaOH and incubating at 98°C for 20 min. Reverse transcribed products were resolved on 10% TBE Urea gels, gel extracted, and isopropanol precipitated. Samples were then circularized using CircLigase (Epicentre) at 60°C for 1 hr, and circularized product was used as template for PCR amplification using primers AATGATACGGCGACCACCGAGATCTACAC and CAAGCAGAAGACGGCATAC-GAGATNNNNNNGTGACTGGAGTTCAGACGTGTGCTCTTCCG, where NNNNNN refers to a six nt barcode. PCR amplification was done using 8–12 cycles using Phusion polymerase (NEB). PCR products were resolved on 10% TBE gels, gel extracted, and then precipitated using isopropanol. PCR products were analyzed for size and concentration using a BioAnalyzer high sensitivity DNA kit before sequencing on an Illumina HiSeq 2500.

## TCA amino acid incorporation assay

Frozen cell pellets were cryogenically pulverized with the standard lysis buffer with antibiotics or varying salt concentrations as indicated in the text. 20 g of the resulting frozen lysate was thawed and precleared by centrifugation at 20,000 g for 10 min and quantified using the absorbance at A260. [14]C-Lys tRNA was prepared as follows. 5 µM purified tRNA (Chemical Block, Russia) was incubated in 100 mM HEPES-KOH pH 7.6, 10 mM ATP, 1 mM DTT, 10 mM KCl, 20 mM MgCl$_2$, 50 µM $^{14}$C-amino-acid and 1 µM synthetase at 37°C for 30 min. 2 µL (0.5 µM) labeled tRNA and 20 g frozen lysate were mixed and allowed to thaw for 15 min at room temperature. 100 µL 1 M NaOH was added to the reaction and incubated at RT for 20 min. Nascent peptides were precipitated by adding 10% TCA +5% casamino acids and collected on glass microfiber filters (Whatman). Membranes were washed on a vacuum apparatus with 5% TCA and 80% EtOH to remove any free

labeled amino acids. [14C]-Lys incorporation was determined with a scintillation counter for four technical replicates from the same lysate.

## tRNA northern blot analysis of tRNA aminoacylation levels

Cultures were either flash frozen or filtered as described above. From 30 g of flash frozen culture, total RNA was extracted using 15 mL Trizol (Invitrogen), 3 mL 3 M NaOAc pH 5.0 and 30 µL 1 M EDTA. The samples were then gently vortexed for 5 min and centrifuged for 10 min at 8000 g. Following an additional wash with 10 mL acid phenol pH 4.5, the aqueous layer was ethanol precipitated. From filtered cultures, total RNA from the frozen pellet was extracted with 500 µL of 0.3 M NaOAc pH 5.0, 1 mM EDTA, and 500 µL Trizol. Following an additional wash with 500 µL acid phenol pH 4.5, the aqueous layer was ethanol precipitated. From each samples, 1.5 µg RNA was deacylated in 0.2 M Tris pH 9 at 37°C for 2 hr and ethanol precipitated. Untreated and deacylated RNA were then oxidized using sodium periodate (5 mM $NaIO_4$, 50 mM NaOAc pH 5.0) for 60 min at 37°C; glucose was added to a final concentration of 50 mM and the RNA was ethanol precipitated. 500 µL 1 M lysine pH 8.5 was added to promote ß-elimination of oxidized 3′ RNA ends. Samples were then purified by an acid phenol/chloroform extraction and ethanol precipitation. Samples were run on a 11% TBE 7 M Urea denaturing polyacrylamide gel. RNA was transferred using a semi-dry transfer apparatus (Biorad) onto a Zeta-Probe nylon membrane. RNA was UV crosslinked to the membrane using 3600 µJ UV light (UV Stratalinker 2400). Membranes were probed with 5′-32P GATTCGAACTCTGGAACCCTTTCGGGTCGCCGGTTTTC (tRNA$^{Ser}$ UGA), 5′-32P GAATCGAACCCGCATCATCAGCTTGG (tRNA$^{Gly}$ UCC), or 5′-32P GACTTGAACCCCCACGTCCGTAAGGACACTAACACCTG (tRNA$^{Leu}$ CAG) and signal was detected on a Typhoon phosphorimager.

## Analysis of ribosome profiling data

Custom Python scripts were used to analyze sequencing data in iPython notebook (*Mohammad, 2018*). Raw reads were filtered for quality and trimmed using Skewer v0.2.2. Bowtie v0.12.7 was used to map reads uniquely to genome build NC_000913.2 (allowing two mismatches) after reads mapping to tRNA or rRNA were discarded. Ribosome density was assigned to the 3′-end of reads using read sizes 10–40 nt in *Figures 1* and *3*. We found that in libraries where the ribosomes were pelleted prior to nuclease digestion, MNase cleaves mRNA within the ribosome, presumably because tRNAs are depleted. As a result, the 3′-ends of short RPFs are shifted compared to RPFs that span the whole ribosome, as shown for start codon peaks and Ser codons in L27 in *Figure 5— figure supplement 1E and F*. In analyzing these libraries in *Figures 5–7*, we used RPFs > 23 nt to ensure that all the 3′-ends are properly aligned. Genes with fewer than 0.5 reads per nucleotide on average were excluded from analysis. On each gene, codons close to the ends of the gene were likewise excluded (27 nt downstream of the start codon and 12 nt upstream of the stop codon).

To calculate pause scores we normalized the read count at each nt of a gene by dividing by the mean read count for the gene. For each codon, we calculated the mean value including reads from all three nt. Average pause scores were calculated using these values from all instances of the codon or amino acid of interest. Pause scores calculated for the A site used a −11 nt shift; P- and E-site pause scores used a shift of −14 and −17, respectively. Tripeptide pause scores were calculated with the last of the three codons in the A site. Asymmetry scores were calculated as the log2 value of the ratio of total density on the second half of a gene over the total density on the first half.

## Data availability

The sequencing data reported in this publication have been deposited in NCBI's Gene Expression Omnibus and are available through GEO Series accession number GSE119104 (https://www.ncbi.nlm.nih.gov/geo/query/acc.cgi?acc=GSE119104). Custom Python scripts (*Mohammad, 2018*) and the iPython notebook used to analyze the data are available at https://github.com/greenlabjhmi/2018_Bacterial_Pipeline_riboseq (copy archived at https://github.com/elifesciences-publications/2018_Bacterial_Pipeline_riboseq).

## Acknowledgments

The authors thank Boris Zinshteyn, Colin Wu, and Jeff Hussman for their insights into the ribosome profiling protocol and David Mohr at the Genetics Resources Core Facility, Johns Hopkins Institute

of Genetic Medicine for sequencing assistance. This study was funded by NIH grant GM110113 to AB, by the Protein Translation Research Network (GM105816), and by HHMI (RG).

# Additional information

### Competing interests
Rachel Green: Reviewing editor, *eLife*. The other authors declare that no competing interests exist.

### Funding

| Funder | Grant reference number | Author |
|---|---|---|
| National Institute of General Medical Sciences | GM110113 | Allen R Buskirk |
| Howard Hughes Medical Institute | | Rachel Green |
| National Institute of General Medical Sciences | GM105816 | Allen R Buskirk |

The funders had no role in study design, data collection and interpretation, or the decision to submit the work for publication.

### Author contributions
Fuad Mohammad, Formal analysis, Investigation, Methodology; Rachel Green, Conceptualization, Funding acquisition, Writing—review and editing; Allen R Buskirk, Conceptualization, Formal analysis, Supervision, Funding acquisition, Writing—original draft, Project administration

### Author ORCIDs
Rachel Green (iD) http://orcid.org/0000-0001-9337-2003
Allen R Buskirk (iD) http://orcid.org/0000-0003-2720-6896

### Decision letter and Author response
Decision letter https://doi.org/10.7554/eLife.42591.037
Author response https://doi.org/10.7554/eLife.42591.038

# Additional files

### Supplementary files
• Transparent reporting form
DOI: https://doi.org/10.7554/eLife.42591.013

### Data availability
Sequencing data have been deposited in GEO under accession code GSE119104. Custom Python scripts and the iPython notebook used to analyze the data are available at https://github.com/green-labjhmi/2018_Bacterial_Pipeline_riboseq (copy archived at https://github.com/elifesciences-publications/2018_Bacterial_Pipeline_riboseq).

The following dataset was generated:

| Author(s) | Year | Dataset title | Dataset URL | Database and Identifier |
|---|---|---|---|---|
| Mohammad F | 2018 | A systematically-revised ribosome profiling method for bacteria reveals pauses at single-codon resolution | https://www.ncbi.nlm.nih.gov/geo/query/acc.cgi?acc=GSE119104 | NCBI Gene Expression Omnibus, GSE119104 |

The following previously published datasets were used:

| Author(s) | Year | Dataset title | Dataset URL | Database and Identifier |
|---|---|---|---|---|
| Marks JP, Kannan K, Roncase E, Orelle C, Kefi A, Klepacki D, Vázquez-Laslop N, Mankin AS, Marks JP, Kannan K, Roncase E, Orelle C, Kefi A, Klepacki D, Vázquez-Laslop N, Mankin AS | 2016 | Context-specific inhibition of translation by ribosomal antibiotics targeting the peptidyl transferase center | https://www.ncbi.nlm.nih.gov/geo/query/acc.cgi?acc=GSE86536 | NCBI Gene Expression Omnibus, GSE86536 |
| Latif H, Szubin R, Zengler K, Palsson BO | 2015 | A streamlined ribosome profiling protocol for the characterization of microorganisms | https://www.ncbi.nlm.nih.gov/geo/query/acc.cgi?acc=GSE63858 | NCBI Gene Expression Omnibus, GSE63858 |
| Liu X, Jiang H, Gu Z, Roberts JW | 2013 | High-resolution view of bacteriophage lambda gene expression by ribosome profiling | https://www.ncbi.nlm.nih.gov/geo/query/acc.cgi?acc=GSE47509 | NCBI Gene Expression Omnibus, GSE47509 |
| Oh E, Becker AH, Sandikci A, Huber D, Chaba R, Gloge F, Nichols RJ, Typas A, Gross CA, Kramer G, Weissman JS, Bukau B | 2011 | Selective ribosome profiling reveals the cotranslational chaperone action of trigger factor in vivo | https://www.ncbi.nlm.nih.gov/geo/query/acc.cgi?acc=GSE33671 | NCBI Gene Expression Omnibus, GSE33671 |
| Baggett N, Zhang Y, Gross C | 2017 | Global analysis of translation termination in E. coli | https://www.ncbi.nlm.nih.gov/geo/query/acc.cgi?acc=GSE88725 | NCBI Gene Expression Omnibus, GSE88725 |
| Li G, Burkhardt D, Gross CA, Weissman JS | 2014 | Quantifying absolute protein synthesis rates reveals principles underlying allocation of cellular resources | https://www.ncbi.nlm.nih.gov/geo/query/acc.cgi?acc=GSE53767 | NCBI Gene Expression Omnibus, GSE53767 |
| Li G, Oh E, Weissman JS | 2012 | The anti-Shine-Dalgarno sequence drives translational pausing and codon choice in bacteria | https://www.ncbi.nlm.nih.gov/geo/query/acc.cgi?acc=GSE35641 | NCBI Gene Expression Omnibus, GSE35641 |
| Haft RJ, Landick R | 2014 | Correcting direct effects of ethanol on translation and transcription machinery confers ethanol tolerance in bacteria | https://www.ncbi.nlm.nih.gov/geo/query/acc.cgi?acc=GSE56372 | NCBI Gene Expression Omnibus, GSE56372 |
| Subramaniam AR, Zid BM | 2014 | An integrated approach reveals regulatory controls on bacterial translation elongation | https://www.ncbi.nlm.nih.gov/geo/query/acc.cgi?acc=GSE51052 | NCBI Gene Expression Omnibus, GSE51052 |
| Mohammad F, Woolstenhulme CJ, Green R, Buskirk AR | 2016 | Clarifying the Translational Pausing Landscape in Bacteria by Ribosome Profiling | https://www.ncbi.nlm.nih.gov/geo/query/acc.cgi?acc=GSE72899 | NCBI Gene Expression Omnibus, GSE72899 |

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
