## [Decision Letter]

Thank you for submitting your article "A systematically-revised ribosome profiling method for bacteria reveals pauses at single-codon resolution" for consideration by *eLife*. Your article has been reviewed by three peer reviewers, including Alan Hinnebusch as the Reviewing Editor, and the evaluation has been overseen by Kevin Struhl as the Senior Editor. The following individual involved in review of your submission has agreed to reveal his identity: Pavel V Baranov (Reviewer #2).

The reviewers have discussed the reviews with one another and the Reviewing Editor has drafted this decision to help you prepare a revised submission.

This manuscript describes modifications of the ribosome profiling technique for bacteria that enable the identification of elongation pauses at individual codons. They show that the current consensus approach of rapid filtration of cells and addition of chloramphenicol to the lysis buffer during pulverization of frozen cells harvested by rapid filtration is not adequate owing partly to low-level translation that continues in the presence of chloramphenicol (Cm), leading to Cm-induced pauses with Ala, Gly, or Ser codons in the E-site. In addition, they show that rapid filtration leads to reduced charging of serine and glycine tRNAs and attendant pausing at Ser or Gly codons in the A site. To remedy these problems, they show that arresting translation with high concentrations of magnesium in the lysis buffer rather than Cm, and rapid freezing of cells without filtration, eliminates virtually all of the artifactual pausing and yields an A-site pause frequency profile that is inversely correlated with codon-adaptation index (CAI) in the expected manner. They also show that aligning the 3' ends of ribosome protected foot-prints (RPFs) to the genome sequence yields much better triplet-periodicity compared to 5' alignments, and explain that a wide range of protected fragments should be sequenced to avoid biases either for or against RPFs that include anti-Shine-Dalgarno sequence matches at their 5' ends.

All of the reviewers agreed that the work is significant because it should set a new standard for the proper application of ribosome profiling of bacterial cells. However, there are a few issues to be addressed.

Given that the central point of the paper is to show that the revised technique allows unbiased detection of elongation pauses, the authors are asked to examine one additional sequencing library prepared with frozen cells and MgCl_2_ but using RelE in place of MNase, and determine whether this improves the inverse correlation of codon pausing with CAI described in Figure 7, which is currently substantially less than reported by Weinberg et al. for yeast RPFs. It seems important to determine whether the weaker correlation reflects a real biological difference or a technical shortcoming that can be solved by using a more precise nuclease to trim the RPFs.

The authors are also asked to make the appropriate additions or changes in text to address the other comments raised by the reviewers, shown in the separate reviews below.

*Reviewer #1:*

This manuscript describes several modifications of the ribosome profiling technique for bacteria that enable the identification of elongation pauses at individual codons. They show that the current consensus approach of rapid filtration of cells and addition of chloramphenicol to the lysis buffer during pulverization of frozen cells harvested by rapid filtration is not adequate owing partly to low-level translation that continues in the presence of chloramphenicol (Cm), leading to Cm-induced pauses with Ala, Gly, or Ser codons in the E-site. In addition, they show that rapid filtration leads to reduced charging of serine and glycine tRNAs and attendant pausing at Ser or Gly codons in the A site. To remedy these problems, they show that arresting translation with 150 mM MgCl_2_ in the lysis buffer rather than Cm and rapid freezing of cells without filtration eliminates virtually all of the artifactual pausing and yields an A-site pause frequency profile that is inversely correlated with codon-adaptation index (CAI) in the expected manner. They also show that aligning the 3' ends of ribosome protected foot-prints (RPFs) to the genome sequence yields much better triplet-periodicity compared to 5' alignments, and explain that a wide range of protected fragments should be sequenced to avoid biases either for or against RPFs that include anti-Shine-Dalgarno sequence matches at their 5' ends.

This work is significant because it should set the standard for the proper application of ribosome profiling of bacteria. However, there are a few issues to be addressed.

The authors note that even with 3' alignments of reads, the triplet periodicity of the aligned reads is not very good, and actually reflects the sequence bias of MNase cleavage of naked RNA. This leads to two comments.

– First, the way it's written one might infer that a plot of the kind shown in Figure 1A but constructed for total aligned RNA reads would look very similar to that presented here for aligned RPFs. Presumably this is not the case considering that there is no periodicity observed upstream of the AUG start codon. Perhaps they should include such a plot constructed from RNA reads to assure the reader of this point.

– Second, the authors state that "For studies where the reading frame is essential…generating RPFs… with RelE" should be done. I feel that the current study falls into this category and that the authors should examine one additional library prepared with frozen cells and MgCl_2_ but using RelE in place of MNase, and determine whether this improves the inverse correlation of codon pausing with CAI described in Figure 7, which is currently substantially less than observed by Weinberg et al. for yeast RPFs.

*Reviewer #2:*

Ribosome profiling is a powerful technique that not only can be used for detection of translated regions and their differential expression, but also for characterization of general parameters of translation such as relative decoding rates of specific codon as well as fine details of specific mRNA translation. The experimental procedures required by ribosome profiling protocol often affect translation, thus distorting its ability to reflect the real cellular translation accurately. It is important to characterize these distortions and design alternative procedures that minimize such distortions. This manuscript describes the authors' quest towards the perfect ribosome profiling protocol in bacteria.

The authors reported two main findings:

1) Chloramphenicol (like cycloheximide) does not completely block translation even when it is added to the lysates (and not in the media), because translation continues after lysis. This finding is not spectacularly surprising, but it is very important that the authors characterized it.

2) Filtering procedure that is often carried out before liquid nitrogen freezing leads to pauses at Ser and Gly codons. This finding is bizarre. The authors went further and looked at how amino acylation status of tRNAs is affected by the procedure and found that Ser and Gly tRNAs are selectively deacylated during this procedure. This is still bizarre and potentially very interesting. Of course, further exploration of this is outside of the scope of this work, but I hope the authors will follow up on this in a separate work. It is a remarkable example of translation apparatus sensitivity to changes in physiological conditions.

Further, the authors came up with the alternative experimental procedures to eliminate these artefacts. They found that high magnesium concentration (or low Na^+^/K^+^) completely blocks translation and the use of corresponding buffer conditions removes the requirement of translation inhibitors during ribosome profiling. As an alternative to filtration the authors used media dispension directly into liquid nitrogen.

Using the improved procedure, the authors have been able to remove artefactual pauses and found that ribosome footprint density anticorrelates with codon adaptation index as expected theoretically.

In addition to offering an improved ribosome profiling (and being an exemplary work into technical interrogation of the protocol), the manuscript offers a very useful discussion of ribosome profiling procedures that goes beyond what is used in bacteria and, in my opinion, this is a must read for anyone who generates or analyses ribosome profiling data.

*Reviewer #3:*

This manuscript presents a thorough analysis of technical considerations in bacterial ribosome profiling and their impact on biological interpretation of these data. Bacterial ribosome footprints span a broad size range that is only partly explained by the sequence biases of micrococcal nuclease (MNase), the enzyme used to create footprints. Next, it is reported that translation elongation can continue post-lysis in bacterial cell extracts, and a protocol for arresting this translation with high Mg^2+^ is reported. Finally, data are presented arguing that harvesting cells by filtration induces detectable stress responses, and a bulk freezing protocol is described.

This manuscript will advance the study of bacterial translation. The data are very clear, and the manuscript links the specific technical concerns with biological misinterpretations. Many of the concerns raised here were recognized in the first bacterial ribosome profiling studies, but the solutions presented here are valuable and novel.

1) The manuscript mentions unpublished data in a few places. The yeast unpublished data is interesting but largely superfluous to the arguments presented here. The *B. subtilis* profiling data seems more central to the generality of results presented here – does this manuscript describe the best approach for *E. coli* ribosome profiling, or for bacterial ribosome profiling more generally?

2) The manuscript discusses serine deacylation induced stress at length. One aspect of filtration that is often overlooked is the potential for cold shock. Was filtration carried out at 37 ºC with a pre-warmed filtration apparatus? If so, this should be mentioned in the Materials and methods; if not, this should be discussed as a possible stressor.

3) The potential for Cm-mediated distortions was recognized in the original Oh et al. (2011) manuscript. The authors here investigate "L9" and "L10" as Cm-pretreated libraries, but the same manuscript also reported SRR364370 and SRR364368 which were "harvested by rapid filtration".

4) Some of the analysis in the very recent manuscript Zhao et al. (2018) seems relevant to the discussion of footprint length and reading frame analysis.

5) The pattern of pausing seen in the optimized bacterial ribosome profiling data resembles the pattern reported for mammalian ribosomes in Ingolia et al. (2011). This similarity seems remarkable.

---

## [Author Response]

[…] However, there are a few issues to be addressed.Given that the central point of the paper is to show that the revised technique allows unbiased detection of elongation pauses, the authors are asked to examine one additional sequencing library prepared with frozen cells and MgCl_2_ but using RelE in place of MNase, and determine whether this improves the inverse correlation of codon pausing with CAI described in Figure 7, which is currently substantially less than reported by Weinberg et al. for yeast RPFs. It seems important to determine whether the weaker correlation reflects a real biological difference or a technical shortcoming that can be solved by using a more precise nuclease to trim the RPFs.

The authors are also asked to make the appropriate additions or changes in text to address the other comments raised by the reviewers, shown in the separate reviews below.

As pointed out by reviewer #1, we see a weaker inverse correlation between pausing and CAI than is observed in yeast. In the Discussion section (fourth paragraph) we explain our view that this is due to technical shortcomings, not real biological differences. In particular, we know that we are sampling ribosomes in different steps of the elongation cycle: some are arrested during peptidyl transfer (we can see pausing at Pro codons) and some during decoding (we also see pauses at rare codons). These subpopulations pause for different reasons, complicating the signal. In yeast, it is now possible to separate these subpopulations and sample only decoding ribosomes, but this is not yet possible in bacteria. We are working to address this issue but it is beyond the scope of this study; it took several years of optimization in yeast to solve this problem.

The reviewers asked us to use RelE (together with MNase) to generate ribosome footprints to see if this would improve the inverse correlation between pausing and CAI. The problem is that RelE cleaves mRNA at the A-site codon and has strong sequence selectivity. Since we assign ribosome positions from the 3’-end of fragments, this introduces bias at the exact position we want to study, the A-site codon. In contrast, because MNase cleaves at the 3’-boundary of the ribosome, roughly 12 nt away from the A site, the sequence selectivity of MNase creates little or no bias at the A-site codon after averaging instances of a given codon. In short, RelE is great for reading frame but bad for pausing analyses. We changed our discussion of RelE in the Results section to explain these limitations (subsection “On the sequence specificity of nucleases”, last paragraph).

As prompted by the reviewers, we prepared samples with our new methods (direct freezing, high Mg buffers), pelleted the ribosomes, resuspended them in the standard buffer (with low Mg levels), and generated footprints using MNase together with various concentrations of RelE. Even at high concentrations of RelE, 5 times what we previously reported, only a small fraction of reads (15%) were cleaved by RelE. This may be because our protocol traps ribosomes in a conformation that prevents RelE from binding in the A site. Preliminary experiments with high Mg buffers in yeast without any antibiotics yield 28 mer footprints characteristic of an occupied A site (perhaps in the hybrid/rotated state); this would be incompatible with RelE binding.

Reviewer #1:[…] This work is significant because it should set the standard for the proper application of ribosome profiling of bacteria. However, there are a few issues to be addressed.The authors note that even with 3' alignments of reads, the triplet periodicity of the aligned reads is not very good, and actually reflects the sequence bias of MNase cleavage of naked RNA. This leads to two comments.– First, the way it's written one might infer that a plot of the kind shown in Figure 1A but constructed for total aligned RNA reads would look very similar to that presented here for aligned RPFs. Presumably this is not the case considering that there is no periodicity observed upstream of the AUG start codon. Perhaps they should include such a plot constructed from RNA reads to assure the reader of this point.

3 nt periodicity arises from both the sequence selectivity of MNase and the bias of the genetic code in open reading frames. There is no periodicity in untranslated regions because there is no systematic nucleotide bias with 3 nt periodicity in these regions of the genome. In the Hwang and Buskirk (2017) paper referenced here, we prepared RNA-seq libraries with MNase and observed 3 nt periodicity exclusively in ORFs. To clarify this issue in the text, we changed the wording of the second paragraph of the subsection “On the sequence specificity of nucleases”.

– Second, the authors state that "For studies where the reading frame is essential…generating RPFs… with RelE" should be done. I feel that the current study falls into this category and that the authors should examine one additional library prepared with frozen cells and MgCl_2_ but using RelE in place of MNase, and determine whether this improves the inverse correlation of codon pausing with CAI described in Figure 7, which is currently substantially less than observed by Weinberg et al. for yeast RPFs.

See the response regarding RelE above.

Reviewer #3:[…] This manuscript will advance the study of bacterial translation. The data are very clear, and the manuscript links the specific technical concerns with biological misinterpretations. Many of the concerns raised here were recognized in the first bacterial ribosome profiling studies, but the solutions presented here are valuable and novel.1) The manuscript mentions unpublished data in a few places. The yeast unpublished data is interesting but largely superfluous to the arguments presented here. The B. subtilis profiling data seems more central to the generality of results presented here – does this manuscript describe the best approach for E. coli ribosome profiling, or for bacterial ribosome profiling more generally?

The yeast data are now in press at Molecular Cell and a citation has been added to the text. We have not made libraries with our new methods in *B. subtilis*. Our unpublished *B. subtilis* data only show that using RNase I isn’t better than MNase – it doesn’t yield distinct footprint sizes like the 28 mers in yeast. We believe that harvesting cells by filtration and arresting ribosomes with chloramphenicol are generally problematic in bacteria.

2) The manuscript discusses serine deacylation induced stress at length. One aspect of filtration that is often overlooked is the potential for cold shock. Was filtration carried out at 37 ºC with a pre-warmed filtration apparatus? If so, this should be mentioned in the Materials and methods; if not, this should be discussed as a possible stressor.

This is a good question. We added a line to the seventh paragraph of the Discussion, stating that we see strong Ser and Gly pauses whether filtration is performed at room temperature or in a 37 °C room.

3) The potential for Cm-mediated distortions was recognized in the original Oh et al. (2011) manuscript. The authors here investigate "L9" and "L10" as Cm-pretreated libraries, but the same manuscript also reported SRR364370 and SRR364368 which were "harvested by rapid filtration".

We added a line to the last paragraph of the subsection “Chloramphenicol in the media induces artifacts at the gene level”, citing this paper and an associated protocol, both of which address this problem thoughtfully. We believe it is important to address these issues again because people continue to harvest cultures by adding Cm and pelleting cells by centrifugation.

4) Some of the analysis in the very recent manuscript Zhao et al. (2018) seems relevant to the discussion of footprint length and reading frame analysis.

A reference to this paper was added in the new section on nuclease and cloning bias (subsection “On the sequence specificity of nucleases”, first paragraph).

5) The pattern of pausing seen in the optimized bacterial ribosome profiling data resembles the pattern reported for mammalian ribosomes in Ingolia et al. (2011). This similarity seems remarkable.

We added a citation of this paper in this context to the fourth paragraph of the Discussion.